# ShellSet v1.1.0 - Parallel Dynamic Neotectonic Modelling: A case study using Earth5-049

Jon B. May[1], Peter Bird[2,1], and Michele M. C. Carafa[1]

[1]Istituto Nazionale di Geofisica e Vulcanologia (INGV), Sezione di Sismologia e Tettonofisica, L'Aquila, Italy
[2]Department of Earth, Planetary, and Space Sciences, UCLA, Los Angeles, California, U.S.A

**Correspondence:** Jon B. May (jonbryan.may@ingv.it)

**Abstract.**

We present a parallel combination of existing, well known, and robust software used in modelling the neotectonics of planetary lithosphere, which we call ShellSet. An added parallel framework allows multiple models to be run at the same time with varied input parameters. Additionally, we have included a grid search option to automatically generate models within a given parameter space. ShellSet offers significant advantages over the original programs through its simplicity, efficiency, and speed. We demonstrate the performance improvement obtained by ShellSet's parallel framework by presenting timing and speedup information for a parallel grid search, varying the number of processes and models, on both a typical computer and high-performance computing cluster node. A possible use case for ShellSet is shown using two examples in which we improve upon an existing global model. In the first example we improve the model using the same data, in the second example we further improve the model through the addition of a new scoring data set. The necessary ShellSet program version along with all required input and post-processing files necessary to recreate the results presented in this article are available from https://zenodo.org/records/7986808, May et al. (2023).

## 1 Introduction

Recent decades have seen significant progress made in the numerical modelling of lithospheric and crustal-scale processes. Simulations have become increasingly complex, and it often remains difficult to determine the best set of model parameters for a specific simulation, especially if non-linear rheologies or 3D effects are involved. Currently, the determination of the best set of model parameters is usually done by iteratively changing the model input parameters, performing numerous simulations, and scoring them against a particular class of observations, for example GPS measurements; stress tensor orientations; or earthquake production rates.

There has been a significant improvement in computational power, efficiency, and machine availability since the first computers were utilised for scientific purposes. Research institutions, both academic and industrial (public and private), provide employees with personal machines for their work. Even the least powerful of these machines is typically capable of simulations, in serial and parallel, that would have required much more expensive hardware only a decade previously. This allows many

more simulations to be performed within any given period, and larger models to be tested, which means scientific knowledge is being improved almost by the second. In short, current day scientists have access to more computing power than ever.

Unfortunately, this rapid improvement in computing power has, in some cases, outstripped the ability of widely used, scientifically robust, programs to efficiently use the hardware available. For example, the programs Plasti, Ellipsis3D and ConMan, see Fuller et al. (2006); Moresi et al. (2007); King et al. (2020) respectively, all remain serial. While others, for example SNAC and Gale (Choi et al. (2008); Moresi et al. (2012) respectively), solve models in parallel using the Message Passing Interface (MPI) to distribute work across multiple cores, however they solve one model at a time requiring the user to update input parameters either between queued models or launch multiple models in parallel manually. Whilst this parallelisation of the program will yield a performance benefit, should a user wish to perform multiple models a manual procedure is not the most efficient - especially if a methodological set of models is required, for example when searching for an optimal value within a particular parameter or set of parameters.

Shells is another program which, while it can solve a single model in parallel, requires a set of models be controlled manually. Shells is a dynamic neotectonic modelling program first developed and released in 1995, see Kong and Bird (1995), since which it has had several updates, the most recent being released in 2019. Shells uses Intel MKL to solve systems of linear equations in parallel which means that, unlike previously noted parallel software which use MPI, Shells uses OpenMP style threads to parallelise its model. This limits the number of tasks possible in parallel as these threads are limited to a single shared-memory compute device whereas MPI can span a distributed memory machine. Due to this Shells is unable to fully utilise the currently available computing hardware, while also being unable to automatically search for an optimal value/set of values for input parameter(s).

In this work we present ShellSet, which is composed of (i) OrbData5, a program used to alter the input finite element grid (FEG) file which we will refer to as OrbData; (ii) Shells, a dynamic neotectonic modelling program using finite elements and (iii) OrbScore2, a scoring program which we will refer to as OrbScore. A brief explanation of the functionality of each can be found in Sect. 2.1, Sect. 2.2 and Sect. 2.3 respectively. Both OrbData and OrbScore function in conjunction with Shells by design. The input FEG file can be created using a separate program OrbWin maintained by, and available from, the author of Shells, OrbData and OrbScore.

Our program is based entirely on freely-distributed software with OrbData, Shells and OrbScore all being available at: http://peterbird.name/index.htm. All program dependencies: Intel Fortran compiler; Intel Math Kernel Library (MKL); and a Message Passing Interface environment (MPI) are available from Intel within the Base and HPC oneAPI toolkits. The final program runs in a Linux OS environment either using a guest Linux OS on Windows machines (e.g. WSL2) or directly on Linux OS machines, including high-performance computing (HPC) structures. It requires Python3 for the use of its (optional) graphical user interface (GUI) and scatter plotter routines.

This work serves as an introduction to ShellSet, an MPI parallel combination of existing, well known, robust and widely used software, to the neotectonic modelling community. The version of ShellSet used for this article, along with all input files, examples and user documentation is available online, see https://zenodo.org/records/7986808, May et al. (2023). In the

following work we will use the term *model* to refer to a single model (defined by its input parameters) while *test* refers to a set of models.

## 2   Software

At the base of ShellSet are three existing programs: OrbData, Shells, and OrbScore. Despite recent updates, these three programs remain both serial and separate by design. This requires the user to manually control all files required by each program, which may be complicated further depending on the settings for Shells simulations. For example, previously generated output is often required to be read before the next Shells solution iteration begins. We outline the function of each of these programs in the following sections.

### 2.1   OrbData

Before deformation of the lithosphere can be modelled its present structure must be defined, including its surface elevations; layer (crust and mantle-lithosphere) thicknesses; densities; and parameters which define its internal temperatures. These quantities must be specified for each node in the finite element grid which defines the domain of the model. OrbData calculates this lithosphere structure, which is used as an input to both Shells and OrbScore, by simple local operations on published data sets, often in spatially gridded formats. OrbData does not affect the topology of the finite element grid or its included fault network.

OrbData computes the crustal and mantle lithosphere thicknesses based on assumptions of local isostasy and either: (a) a steady state geotherm assumption; or (b) seismically determined layer thicknesses. It incorporates seismic constraints by adding two adjustable parameters, the density anomaly of the lithosphere of compositional origin and an extra quadratic curvature of geotherm due to transient cooling/heating. These extra values are then incorporated into the finite element grid at each node. For a description of the various algorithms OrbData uses, see Bird et al. (2008).

Whether the finite element grid needs updating by OrbData depends on which input variables have been altered. We therefore consider OrbData to be *optional* - it is strictly required in some cases but unnecessary in others. ShellSet removes this consideration from the user by automatically deciding whether or not a certain parameter change requires an update to the finite element grid; this decision is hard coded into the program and is based on which input parameters are altered within a test.

### 2.2   Shells

Shells is the leading program among those who want to conduct physics-based simulations of planetary tectonics with the efficiency of 2D spherical FE grids. Most competing programs use 3D grids, for example Ellipsis3D, SNAC, CitComCU and CitComS (Moresi et al. (2007); Choi et al. (2008); Moresi et al. (2009, 2014) respectively), while Gale (Moresi et al. (2012)) has 2/3D functionality. The 3D grids will typically require greater computing resources for comparable domain sizes (depending on the mesh size), and therefore limit the number of experiments that are practical in parallel.

Shells uses the thermal and compositional structure of thin spherical shells (usually called "plates") of planetary lithosphere, together with the physics of quasi-static creeping flow, to predict patterns of velocity, straining, and fault-slip on the surface of a planet. Since the first publications outlining the first release version, Kong and Bird (1995) and Bird (1998), Shells has been updated and used in numerous publications including Liu and Bird (2002a, b); Bird et al. (2006, 2008); Kalbas et al. (2008); Stamps et al. (2010); Jian-jian et al. (2010); Austermann et al. (2011); Carafa et al. (2015), and more recently Tunini et al. (2017). A primary goal has been to understand the balance of forces that move the plates while a secondary goal is to predict fault slip rates and distributed strain rates for seismic hazard estimation.

Shells is serial in its application except in two parts of its calculations where it makes use of the thread safe Intel MKL routines *dgbsv* and *dgesv*. These routines apply OpenMP type threading within their execution if the problem size is large enough. The number of parallel threads used within each MKL routine is decided at run time by the routines, although it will default to the number of physical cores, environment variables can be altered to specify a strict number or to allow a dynamic selection of threads for MKL routines.

Shells produces six testable predicted fields in each model: relative velocities of geodetic benchmarks; most-compressive horizontal principal stress azimuths; long-term fault heave and throw rates; rates of seafloor spreading; the distribution of seismicity on the map; and fast-polarization directions of split SKS arrivals. Each of these predictions can be scored against observed data.

Shells contains the possibility of defining lateral variations in lithospheric rheology within the model by updating the FEG file using an additional numeric ID for groups of elements. This ID number should correspond to a value within a separate file which defines the values of: the 4 input parameters which describe frictional rheology at low temperature, including an option for lower effective friction in fault elements; the 9 describing the dislocation-creep flow-laws of the crust and mantle-lithosphere layers of the lithosphere. All elements placed within one of these groups will take their values from this new file, only elements within the default group (ID=0) will take values from the standard parameter input file.

## 2.3 OrbScore

OrbScore is used to score any of the six testable Shells predictions, for relative realism, against supplied real data. These six options are: seafloor spreading rates (SSR); geodetic velocities (GV); most compressive horizontal principal stress directions (SD); fast-polarization azimuths of split SKS waves (SA); smoothed seismicity correlation (SC); and fault slip rates (FSR). Information on SSR, GV, SD and SA datasets can be found in Bird et al. (2008) sections 6.1, 6.2, 6.3 and 6.4 respectively. The FSR dataset has been developed recently and information on the dataset generation can be seen in Appendix A. The SC scoring dataset (not used in this work) is taken from the global centroid-moment-tensor (GCMT) catalogue. The basic methodology and approach used in that project are outlined in Dziewonski et al. (1981), with the most recent description of the analyses, including some significant improvements, described in Ekström et al. (2012).

Table 1 lists the 6 misfit scoring options along with the type of error reported, their units, and the name of the file containing the dataset provided within the ShellSet package. Using one, or a combination, of these calculated misfit scores it is possible to

compare different models and perform a tuning of the input parameters to obtain the best misfit score for a particular scoring setup. The best misfit score then provides the optimal values for the tested variables.

**Table 1.** List of misfit scoring options available in ShellSet

| Error | Type | Units | Filename in package |
|-------|------|-------|---------------------|
| SSR | Mean | mm a$^{-1}$ | magnetic_PB2002.dat |
| GV | RMS | mm a$^{-1}$ | GPS2006_selected_subset.gps |
| SD | Mean | deg. | robust_interpolated_stress_for_OrbScore2.dat |
| SA | Mean | deg. | Fouch_2004_SKS_splitting-selected.dat |
| SC | Correlation coefficient | $\times$ | GCMT_shallow_m5p7_1977-2017.eqc |
| FSR | RMS | mm a$^{-1}$ | aggregated_offset_rates.dig |

SSR - seafloor spreading rates, GV - geodetic velocity, SD - most compressive horizontal stress directions, SA - fast-polarization azimuths of split SKS waves, SC - smoothed seismicity correlation, FSR - fault slip rates. SSR, GV, SD, SA & FSR are misfits, SC is a score. The type of score reported for each error can be changed by altering a single line of the OrbScore source code within ShellSet.

In this work, as in Bird et al. (2008), we use a geometric mean of misfits to grade each model. The geometric mean is defined as the $n^{th}$ root of the product of $n$ (assumed positive) numbers:

$$GM = \sqrt[n]{S_1 \ S_2 \ ... \ S_n} \tag{1}$$

This is preferred to any arithmetic mean because the ranking of models using the geometric mean is independent of the units of the ingredient misfits.

## 3 ShellSet

Since the three individual programs are quick in their current form, and are intrinsically linked in their work, we focused this work on combining them into a single entity. Previously, in order to perform multiple models in parallel, the user would have to start multiple models in separate instances and control each one individually. In ShellSet the program controls the run for each, running multiple in parallel if the compute resources exist. This simplifies the user interface while simultaneously leveraging parallel computing to obtain multiple results at the same time. This streamlined process has multiple benefits for the user and their work when compared with the three underlying programs. We will now outline some of the most important additions to the program and state the improvements that these have yielded, and which ShellSet offers over the original separate programs.

### 3.1 Grid Search

We have programmed two options for the alteration of multiple variables in parallel: a list input and a grid search. Each of these options requires one simple input file to be completed which defines: the variables being tested and their values, or their ranges, for the list input or grid search respectively. While other parameter space sampling methods exist, see Reuber (2021) for an

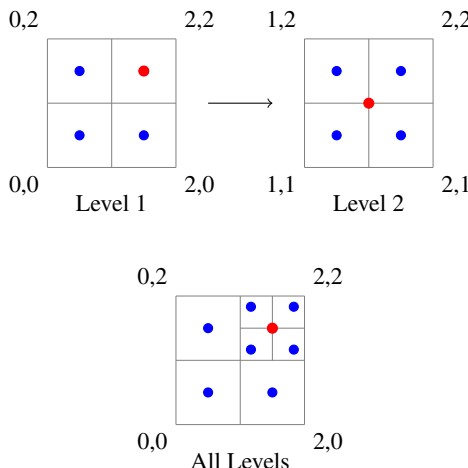

**Figure 1.** Example 2 level grid search that selects a best cell (red point) and divides that cell into 4 new cells.

analysis of various options, we chose to add a grid search as it is simple to program and understand while offering consistent information across the entire parameter space. This means that, should only a simple visualisation of the misfit scores over a parameter space be required, then a single level grid search would provide total coverage with consistent spacing while, for example, a random search would not necessarily provide consistent detail over the entire space.

A grid search is a searching algorithm used to tune parameters towards optimum values within a defined N-dimensional
parameter space. Multiple different schemes can be used within a grid search for how the domain is divided, how to pass the best models to the next level etc, however we use the most basic version. It works by partitioning the parameter space into a grid of equal N-dimensional cells, each cell is represented by a single central model whose location defines the value for each of the tested parameters. The upper left side image of Fig. 1 shows an initial 2×2 grid in a 2-dimensional space, with each of the 4 models represented by a coloured point.

After a full grid of models has been simulated, the algorithm will select a defined number of best models to continue onto the next level. Each of the selected models has its cell divided in the same fashion as the first level (to scale) and representative models at the centre of each cell. In this way the algorithm progressively reduces the size of the searched parameter space by reducing the size of the cells, dividing increasingly smaller parameter spaces into cells and selecting from their best models for further analysis. A typical algorithm will continue in this way until a defined stopping criteria is met, such as a specified
number of levels, a desired model score, a total number of models, etc.

Figure 1 shows the transition from a single best model to a new level. The red point in the upper left image represents the best model, which is selected for the next level shown in the upper right image of the image. The lower image of Fig. 1 shows an overview of these two levels.

The base grid search algorithm was adapted from the Fortran grid search version available from May (2022). Like the
160 original, the grid search used in ShellSet has no limit on the number of dimensions it is possible to use.

In ShellSet we rank the models using their misfit scores, before selecting the best models and refining the search area within the parameter space. Since each model, at each level, is independent an automated search of the parameter space may be performed by the program in parallel, see Sect. 3.2. This parallel search of a parameter space finds optimal values for desired variables quickly (see Sect. 5), efficiently, and allows the program to search each individual parameter to degrees of accuracy that are useful to the user but ordinarily tedious to perform manually.

ShellSet's grid search will work to a defined number of levels whereupon the program will stop. It also optionally allows the user to define a target misfit score, when any model reaches or surpasses this value the program will stop at the completion of that level.

The inclusion of a search algorithm necessitated an additional set of checks to be performed on combinations of input parameters. Originally the programs performed only basic checks (unrealistic value entries on individual parameters, etc.) as a human user was assumed to be in control of all input values. The grid search option essentially replaces the human user with a machine user, which theoretically allows for unrealistic selections of values for input variables. For example, in a test including both fault and continuum friction the fault friction must be less than the continuum friction. This is something which a user would ordinarily handle manually but the grid search algorithm must be controlled using additional checks to prevent unrealistic models being performed. New checks are added into a separated source code file to simplify future updates. We currently place only the most general conditions on some variables (fault friction must be less than continuum friction and the mean mantle density must be greater than that of the crust) and any user is advised to add any conditions needed depending on their local, or global, model requirements to this file in order to keep these conditions in a single location.

## 3.2 ShellSet parallelism

The grid search algorithm is made considerably more efficient if multiple models at the same level are run in parallel. For example, looking at Fig. 1 we can see that at the first level there are four distinct models which could all be performed in parallel, at the second level there are another four. Knowing this we have placed ShellSet into a simple MPI framework which allows the running of multiple models in parallel.

Using MPI to perform multiple models in parallel, ShellSet also maintains the parallelism inherent in the Intel MKL libraries. Therefore, each model run by an MPI process is able to leverage a team of MKL threads within its call to the *dgbsv* and *dgesv* routines if there are available computing resources.

By default the Intel MKL library routines will set the number of threads to a value based on the problem size and number of available cores at run time. This behaviour can be problematic when running multiple models in parallel using MPI, and although it can be changed by altering the program source code (requiring compilation to take effect), we have added a simple control on the maximum number of threads which may be used by MKL routines. Results shown in Sect. 5.1 demonstrate why allowing the MKL routines to select the number of threads automatically is not always optimal - particularly when the possibility of running multiple models in parallel exists. ShellSet calculates the number of free cores available per MPI process at run time and sets the maximum number of threads available to MKL routines to this value. This calculation does not require any extra information from the user.

The migration of file control to the program was necessary to run multiple models in parallel without constant user input, this is discussed in more detail in Sect. 3.3.

### 3.3 User interface

As noted previously, some design features which were necessary in order to perform multiple models in parallel have also caused some *unintended* usability improvements. These, along with designed usability improvements present at its interface
with the user, both in terms of input to the program and the output from it, will now be discussed.

The main output of ShellSet is a single file which defines the command line arguments used to start the program along with a list of results, in which a row represents a model. For every model the altered variables (those defined in the grid search or list input files) are recorded, along with all calculated misfit results, a global model ID number and the ID of the MPI process which performed the model. The two ID numbers allow the user to trace the OrbData, Shells and OrbScore output files related to that
model for further analysis. While this output file was not necessary when manually controlling a set of models its addition in ShellSet gives an immediate *general* understanding of the test results while reporting information required to perform a deeper examination of selected models. This is possible as all output files from OrbData, Shells and OrbScore are saved into private directories with locations and filenames containing the process ID, model number and Shells iteration number.

The simplified input and initialisation user interface has three parts: 1) an input file for the model generation option (list or
grid search); 2) an input file containing a list of input filenames for each of the original programs; 3) a selection of command line arguments which provide run-time information to ShellSet. These are visible in the three red boxes within the dashed olive box labelled *GUI* of Fig. 2. The input file names for each of OrbData, Shells and OrbScore are stored in the input file list which combines with the invocation line and a model input file to initialise a ShellSet test. These steps, while providing the same input required by the individual programs, remove the user's control over any files that are at the interface between OrbData, Shells
and OrbScore, as well as between iterated Shells runs, removing the possibility of errors or delay after program initiation.

Certain experiments require the iteration of the program Shells, with each subsequent iteration requiring a file generated by the previous one. ShellSet is able to handle this for the user. The iteration of Shells is not exact in how many iterations could be required, for example in Bird et al. (2008) 3-5 iterations were used depending on the behaviour of the result after each iteration. Due to this we have added an option to exit from the Shells iteration loop early. This exit may be performed if any
two consecutive Shells iterations have misfit scores that are equal within a preselected tolerance. If activated, ShellSet expects a decimal input at runtime which represents the percentage change from the previous iteration (e.g. 10 % = 0.1), this is used to generate a range from the result of the previous model inside which the new model is deemed *close enough*. Since this is an optional feature there is no default value and the user is expected to provide one when activating it, a *good* value will depend on any requirement for subsequent results to be similar, on the scoring dataset used, and on the underlying problem. The ability
to exit this iteration loop early has the potential to save a lot of time over an entire test. For example, if all 18 models of Bird et al. (2008) required 5 iterations then that totals 90 Shells runs but if each required 3 it would only be 54 runs - that is a saving of 40 % on model numbers.

Along with this option there are eight further command line arguments which allow the user to personalise the test at run time. These eight arguments allow the user to: define the maximum iterations of Shells within the iterative loop; choose between the List and Grid input options; name a new directory where all IO files will be stored; optionally set ShellSet to abort if any *ignorable* error is detected - e.g. by default if any FEG node exists outside of a scoring data input grid then ShellSet will extrapolate the required value without calculation error but report an *ignorable* error, activating this option will cause this issue to abort the program; optionally produce extra program information in a new output file; select the misfit type used to select best models; create a special output file to store models with misfit scores better than a specified value; optionally specify a misfit score at which the grid search algorithm will stop.

ShellSet also contains a geometric mean of misfit scoring option which can be used to rank models. This geometric mean, which users of OrbScore would have to calculate manually after each model has completed, can be comprised of any set of the five misfits generated: SSR, GV, SD, SA, and FSR. Since SC is a score (larger is better) and not a misfit (smaller is better) it should not be included inside this geometric mean. See Table 1 for information on the error types and Eq. (1) for the definition of the geometric mean.

By default the geometric mean is always calculated by ShellSet using all of any of the five noted misfits which are calculated and reported in the output file. It may also be used to rank the models within the grid search by using the relevant command line argument at run time and defining which misfits to use in its calculation.

Included within the ShellSet package, but separate from its main work, there are two new Python programs which improve the usability of the program when compared to the standalone originals. First, we include a simple GUI which helps the user to create/update key input files and select run-time options before launching ShellSet. The GUI helps the user to update the files related to parameter input values, the input file list, and the list and grid input files. It also aids in the creation of a suitable invocation line, see the box labelled *GUI* in Fig. 2. It contains checks on some typical errors for both file editing and program initialisation. The GUI dramatically simplifies the program setup and launching for new users while more experienced users can choose to manually edit the input files using any text editor before launching from the terminal as with Shells. Second, we add a plotting routine that can generate 1D, 2D or 3D scatter plots from the main ShellSet output file. Plots generated by this routine can be seen in Fig. 3 and Fig. 5.

This combination into a single program and the aforementioned updates have reduced errors related to manual control of the original individual programs. While the ability to run models in parallel has saved us and our collaborators valuable research time, allowing the exploration of multiple research hypotheses in a shorter time. Performance analyses using two different compute machines are shown in Sect. 5.2.

## 4   Real world examples

We now present two real world example uses of ShellSet and its grid search option. Both examples follow the same path as in Bird et al. (2008) and search a similar 2D parameter space. All files for both examples are included with the ShellSet package download, see *code and data availability*.

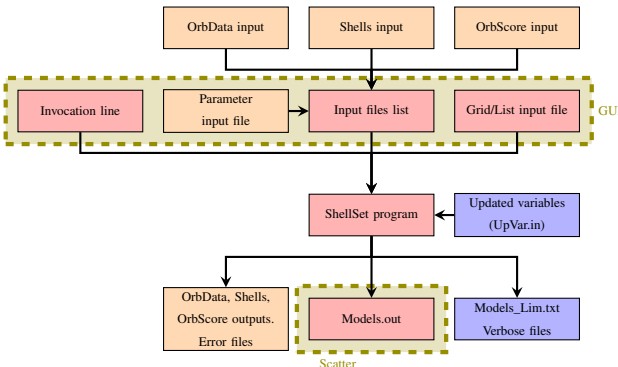

**Figure 2.** Simplified ShellSet schematic. Optional IO is shown in blue boxes, orange boxes show IO from the original programs while red boxes show new to ShellSet functionality. Optional output is switched on at runtime while optional input is switched on by its presence. The parts shown in the shaded, dashed box labelled *GUI* can be handled through the provided GUI, only the Models.out file may be plotted using the provided Python scatter plot routine.

The first example is a direct comparison between the original results, see Table 2 of Bird et al. (2008), and our automated search. In the second example we have added an extra data set used to score the models that was not available to the authors of Bird et al. (2008). While the authors of Bird et al. (2008) had to manually update the parameters within the input files to run their simulations; we will use ShellSet's grid search to search a comparable parameter space that we define as [0.025, 0.8] for

fFric and [1E12, 1E13] for tauMax (N m$^{-1}$).

In the first example the geometric mean of misfits, Eq. (1), is comprised of the SSR, GV, SD and SA datasets as in Bird et al. (2008). The second example expands the geometric mean of misfits to include the new FSR dataset. Information on all misfit types and the geometric mean can be found in Sect. 2.3.

Due to changes in software since 2008 we expect different results to be calculated for the same models. Specifically, there

have been minor updates to the original software and third-party libraries since the publication which will affect its results while there have also been numerous updates to the compilers used in the intervening years. Furthermore, ShellSet is compiled to run on Linux OS whereas the original results were obtained on a version compiled to run on Windows OS. These differences will lead to, for example, different handling of variable precision at minor decimal places which, over multiple iterations, will compound to alter model results in minor but noticeable ways. In order to account for these differences we have rerun the

previous best model, see model Earth5-049 in Table 2 of Bird et al. (2008), and reported the updated results in each example named *New Earth5-049*.

Preliminary testing showed that the requirement of relative velocity convergence to within 0.0001 was too strict for ShellSet; this is likely due to the aforementioned software differences. This gave us choices regarding the linear system solution iterations. We could alter the tolerance limit, increase number of allowed iterations from 50, or both. We decided to loosen the

tolerance to 0.0005 while keeping the iterations set to 50 in order to complete all models in a reasonable time.

As in Bird et al. (2008) each model begins with a trHMax limit of 0.0 Pa for the first call of Shells before updating the limit to $2 \times 10^7$ Pa. Parameter trHMax defines the upper limit of basal shear tractions, so an initial value of 0.0 means that the first iteration of Shells imposes no basal shear tractions.

Detailed analysis of the results is not within the scope of this work, however for each example we will offer some observations and a basic analysis.

## 4.1 Using ShellSet to recreate Bird et al. 2008

In this first example we show a recreation of the work done by the authors of Bird et al. (2008) using ShellSet.

Our grid search was set up to perform 4 iterations of Shells before a final call with updated boundary conditions for a total of 5 Shells iterations per model. After the final run, OrbScore was used to generate misfit scores and the geometric mean of misfits for each model, which was used to rank the models. The geometric mean of misfits, from Eq. (1), for this example is given as

$$GM = \sqrt[4]{S_{SSR}\ S_{GV}\ S_{SD}\ S_{SA}} \tag{2}$$

The grid search partitioned the domain into 9 cells, in a 3×3 grid, each represented by its central model. The best 2 models of these 9 were then chosen to continue to the second level where each represented cell is further divided into a 3×3 grid. This was repeated with the 18 models at this second level to generate another 18 models. This 3-level grid search gives a total of 41 distinct models, once the 4 repeated models are excluded, and a final resolution of 1/27 of the original ranges of $9 \times 10^{12}$ N m$^{-1}$ and 0.775, or $3.\dot{3} \times 10^{11}$ N m$^{-1}$ and 0.0287, for tauMax and fFric respectively.

Table 2 shows some of the results we have obtained using ShellSet. The first two rows show the previous best model, *Earth5-049*, and the scores when this model was rerun, *New Earth5-049*. For brevity we only report the models at the first level, to demonstrate the initial grid (models 1-9), then every model for which the geometric mean of misfits is better than *New Earth5-049*. The entire search history, excluding the first level, can be seen in Table B1 of Appendix B.

It is immediately clear that within the first 9 models the best geometric mean results are obtained with a minimum fFric and tauMax within their respective ranges, with fFric being the more important of the two. This is made clearer in Fig. 3, which maps the entire grid search.

There are eight distinct models (32 being a repetition of model 10) that obtained an equal or better geometric mean of misfits score than *New Earth5-049*. Each one of these eight models achieved a better SSR score than *New Earth5-049*, five of the eight have a better GV score (models 10, 29, 30, 33, 36), none of these eight models have a better score in either SD or SA. The absolute best model is model 29 with a geometric mean of 16.29, an improvement of 0.29.

Examining the full set of models within this test, using Table 2 and Table B1, we can see that over the entire model set no single model has better or equal results in more than two out of the four individual scores that comprise the geometric mean when compared to *New Earth5-049*.

Although a detailed analysis of these results is beyond the scope of this work we have plotted the horizontal velocities of model 29 in Fig. 4 for comparison with Fig. 10 of Bird et al. (2008). We can immediately see that our new model has a lower

**Table 2.** Previous best model of Bird et al. (2008), *Earth5-049*, compared with ShellSet grid search models. *New Earth5-049* is an exact re-run of *Earth5-049* to account for differences in hardware & software. The best model, using the geometric mean score, is shown in bold text.

| Model | fFric | tauMax (N m$^{-1}$) | SSR (mm a$^{-1}$) | GV (mm a$^{-1}$) | SD (deg.) | SA (deg.) | GM |
|---|---|---|---|---|---|---|---|
| Earth5-049 | 0.1 | $2 \times 10^{12}$ | 8.02 | 16.19 | 31.28 | 26.43 | 18.10 |
| New Earth5-049 | 0.1 | $2 \times 10^{12}$ | 7.96 | 12.18 | 30.68 | 25.43 | 16.58 |
| 1 | 0.15417 | $2.5 \times 10^{12}$ | 9.97 | 13.02 | 31.10 | 24.93 | 17.81 |
| 2 | 0.4125 | $2.5 \times 10^{12}$ | 18.84 | 17.43 | 33.82 | 24.90 | 22.93 |
| 3 | 0.67083 | $2.5 \times 10^{12}$ | 26.57 | 21.09 | 35.00 | 25.36 | 26.56 |
| 4 | 0.15417 | $5.5 \times 10^{12}$ | 13.65 | 21.85 | 31.04 | 23.04 | 21.49 |
| 5 | 0.4125 | $5.5 \times 10^{12}$ | 21.90 | 24.11 | 33.43 | 22.28 | 25.04 |
| 6 | 0.67083 | $5.5 \times 10^{12}$ | 28.55 | 24.74 | 34.25 | 22.95 | 27.30 |
| 7 | 0.15417 | $8.5 \times 10^{12}$ | 14.90 | 23.66 | 32.51 | 21.94 | 22.40 |
| 8 | 0.4125 | $8.5 \times 10^{12}$ | 24.02 | 26.29 | 34.15 | 21.31 | 26.03 |
| 9 | 0.67083 | $8.5 \times 10^{12}$ | 30.18 | 27.00 | 35.27 | 22.02 | 28.20 |
| 10 | 0.06806 | $1.5 \times 10^{12}$ | 7.14 | 12.04 | 31.28 | 27.52 | 16.50 |
| 28 | 0.03935 | $1.17 \times 10^{12}$ | 6.57 | 12.27 | 32.00 | 27.65 | 16.34 |
| **29** | **0.06806** | $\mathbf{1.17 \times 10^{12}}$ | **7.03** | **11.70** | **31.57** | **27.10** | **16.29** |
| 30 | 0.09676 | $1.17 \times 10^{12}$ | 7.59 | 11.64 | 31.49 | 26.94 | 16.55 |
| 31 | 0.03935 | $1.5 \times 10^{12}$ | 6.69 | 12.68 | 31.73 | 27.50 | 16.49 |
| 32 | 0.06806 | $1.5 \times 10^{12}$ | 7.14 | 12.04 | 31.28 | 27.52 | 16.50 |
| 33 | 0.09676 | $1.5 \times 10^{12}$ | 7.74 | 11.96 | 31.03 | 26.35 | 16.58 |
| 35 | 0.06806 | $1.83 \times 10^{12}$ | 7.31 | 12.27 | 30.99 | 27.01 | 16.55 |
| 36 | 0.09676 | $1.83 \times 10^{12}$ | 7.86 | 12.09 | 30.87 | 25.66 | 16.56 |

Misfits are reported to 2 decimal places but calculated to greater accuracy. Misfit types are: SSR - seafloor spreading rates, GV - geodetic velocity, SD - most compressive horizontal stress directions, SA - fast-polarization azimuths of split SKS waves. The geometric mean (GM) is calculated as in Eq. (2).

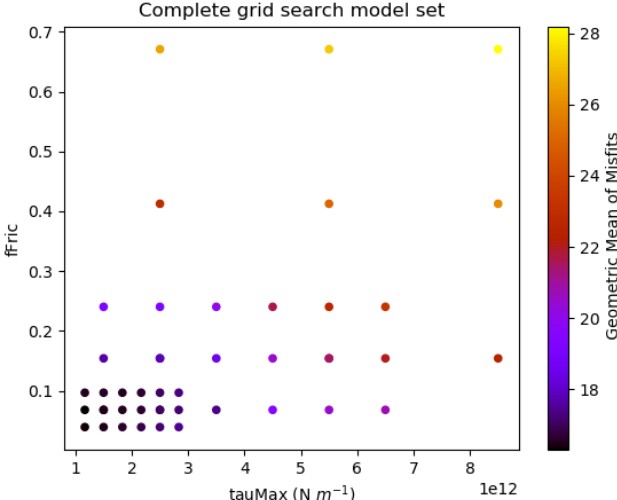

**Figure 3.** Visualisation of the grid search model set generated by ShellSet. The best results are found in the lower left corner of the search area, corresponding to lower values for fFric and tauMax. This plot was generated with ShellSet's included Python scatter plotter.

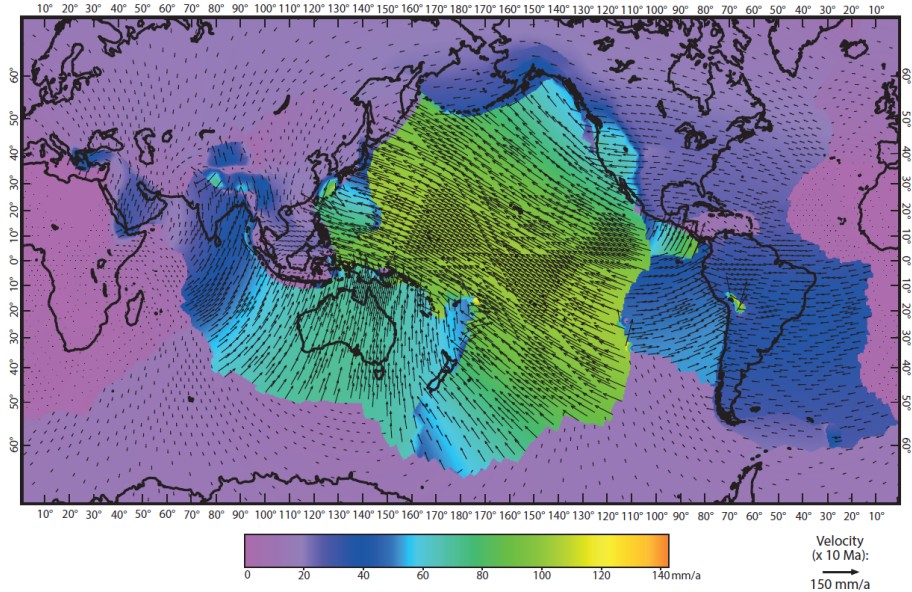

**Figure 4.** Surface horizontal velocity field of model 29.

maximum horizontal velocity than *Earth5-049* with a maximum of a little over 140 mm a$^{-1}$ compared to 153.9 mm a$^{-1}$. Interestingly, we can clearly see an increase in velocities around Nepal from approximately 40 mm a$^{-1}$ to approximately 60 mm a$^{-1}$. With all other inputs being equal between the two models these differences are caused solely by the updated values of fFric and tauMax.

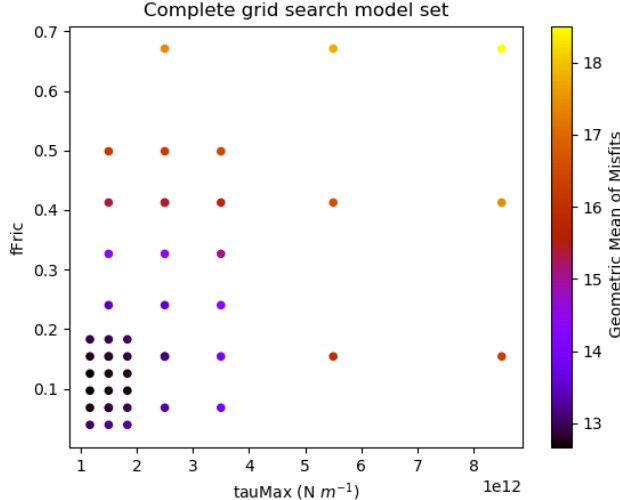

**Figure 5.** Visualisation of the grid search model set generated by ShellSet. The best results are found in the lower left corner of the search area, corresponding to lower values for fFric and tauMax. This plot was generated with ShellSet's included Python scatter plotter.

## 4.2 ShellSet with additional Fault Slip Rate data set

The second example performs the same test as the first example with an added data set that defines the fault slip rate (FSR),
see Appendix A for further information on this data set. We note that the fault slip rate data set was not available to the authors
of Bird et al. (2008).

This additional data set was used to generate an extra misfit for each model, which was then used within the calculation of
the geometric mean of misfits as in Eq. (1), which becomes:

$$GM = \sqrt[5]{S_{SSR} \ S_{GV} \ S_{SD} \ S_{SA} \ S_{FSR}} \tag{3}$$

In the first 2 rows of Table 3 we report the best model of Bird et al. (2008), *Earth5-049*, and the best model of Example 1,
*Ex1 29*. We rerun these two best models adding the new FSR data set to generate fair comparison models, *New Earth5-049* and
*New Ex1 29* respectively.

As in example 1 the search favours lower values for fFric and tauMax however, in this example, the more important variable
is tauMax. This search has generated four models that are an improvement, or equal, to *New Earth5-049* and *New Ex1 29*, see
models 28, 39, 42 and 45, with model 38 being a repeat of *Ex1 29*. The addition of the fault slip rate scoring data set has meant
that every model has a better geometric mean score than in example 1, see Table 2. The absolute best model is model 28 with
a geometric mean that is 0.07 lower than *New Earth5-049* and 0.08 lower than *New Ex1 29*.

Although, as noted, every model run in both examples 1 and 2 has a lower geometric mean score in the second example,
not every reduction is equal and so the alteration to the geometric mean has caused alternate models to be selected between

**Table 3.** Previous best model of Bird et al. (2008), *Earth5-049*, and example 1 compared with ShellSet grid search models. *New Earth5-049* and *New Ex1 29* are exact re-runs of *Earth5-049* and the best model of example 1 respectively. The best model, using the geometric mean score, is shown in bold text.

| Model | fFric | tauMax (N m$^{-1}$) | SSR (mm a$^{-1}$) | GV (mm a$^{-1}$) | SD (deg.) | SA (deg.) | FSR (mm a$^{-1}$) | GM |
|---|---|---|---|---|---|---|---|---|
| Earth5-049 | 0.1 | $2 \times 10^{12}$ | 8.02 | 16.19 | 31.28 | 26.43 | $\times$ | 18.10 |
| Ex1 29 | 0.06806 | $1.17 \times 10^{12}$ | 7.03 | 11.70 | 31.57 | 27.10 | $\times$ | 16.29 |
| New Earth5-049 | 0.1 | $2 \times 10^{12}$ | 7.96 | 12.18 | 30.68 | 25.43 | 4.42 | 12.73 |
| New Ex1 29 | 0.06806 | $1.17 \times 10^{12}$ | 7.03 | 11.70 | 31.57 | 27.10 | 4.78 | 12.74 |
| 1 | 0.15417 | $2.5 \times 10^{12}$ | 9.97 | 13.02 | 31.10 | 24.93 | 3.84 | 13.10 |
| 2 | 0.4125 | $2.5 \times 10^{12}$ | 18.84 | 17.43 | 33.82 | 24.90 | 3.15 | 15.41 |
| 3 | 0.67083 | $2.5 \times 10^{12}$ | 26.57 | 21.09 | 35.00 | 25.36 | 3.22 | 17.41 |
| 4 | 0.15417 | $5.5 \times 10^{12}$ | 13.65 | 21.85 | 31.04 | 23.04 | 4.83 | 15.95 |
| 5 | 0.4125 | $5.5 \times 10^{12}$ | 21.90 | 24.11 | 33.43 | 22.28 | 3.23 | 16.63 |
| 6 | 0.67083 | $5.5 \times 10^{12}$ | 28.55 | 24.74 | 34.25 | 22.95 | 3.20 | 17.78 |
| 7 | 0.15417 | $8.5 \times 10^{12}$ | 14.90 | 23.66 | 32.51 | 21.94 | 4.62 | 16.34 |
| 8 | 0.4125 | $8.5 \times 10^{12}$ | 24.02 | 26.29 | 34.15 | 21.31 | 3.53 | 17.46 |
| 9 | 0.67083 | $8.5 \times 10^{12}$ | 30.18 | 27.00 | 35.27 | 22.02 | 3.43 | 18.51 |
| **28** | **0.12546** | $\mathbf{1.17 \times 10^{12}}$ | **8.46** | **11.85** | **32.06** | **25.78** | **3.93** | **12.66** |
| 38 | 0.06806 | $1.17 \times 10^{12}$ | 7.03 | 11.70 | 31.57 | 27.10 | 4.78 | 12.74 |
| 39 | 0.09676 | $1.17 \times 10^{12}$ | 7.59 | 11.64 | 31.49 | 26.94 | 4.35 | 12.67 |
| 42 | 0.09676 | $1.5 \times 10^{12}$ | 7.74 | 11.96 | 31.03 | 26.35 | 4.39 | 12.72 |
| 45 | 0.09676 | $1.83 \times 10^{12}$ | 7.86 | 12.09 | 30.87 | 25.66 | 4.44 | 12.73 |

Misfits are reported to 2 decimal places but calculated to greater accuracy. Misfit types are: SSR - seafloor spreading rates, GV - geodetic velocity, SD - most compressive horizontal stress directions, SA - fast-polarization azimuths of split SKS waves, FSR - fault slip rate. The geometric mean (GM) is calculated as in Eq. (3).

levels. This alteration takes the grid search through a different path to find its optimal model. We will now outline where the deviations within the search paths occur.

In example 1 the best two models of the first level are models 1 and 4 (see Table 2) while in example 2 the best models are 1 and 2 (see Table 3). Since all other misfit scores remain equal between the two examples this change from selecting model 4 to selecting model 2 is wholly caused by the addition of the fault slip rate data set altering the geometric mean score calculation. The best models at the second level are also, by chance, run in both examples. In the first example these models are 10 and 13 while in the second they are 10 and 11 (see Tables B1 and B2). The best model in the second example is not run in the first, this is an effect of the updated geometric score calculation.

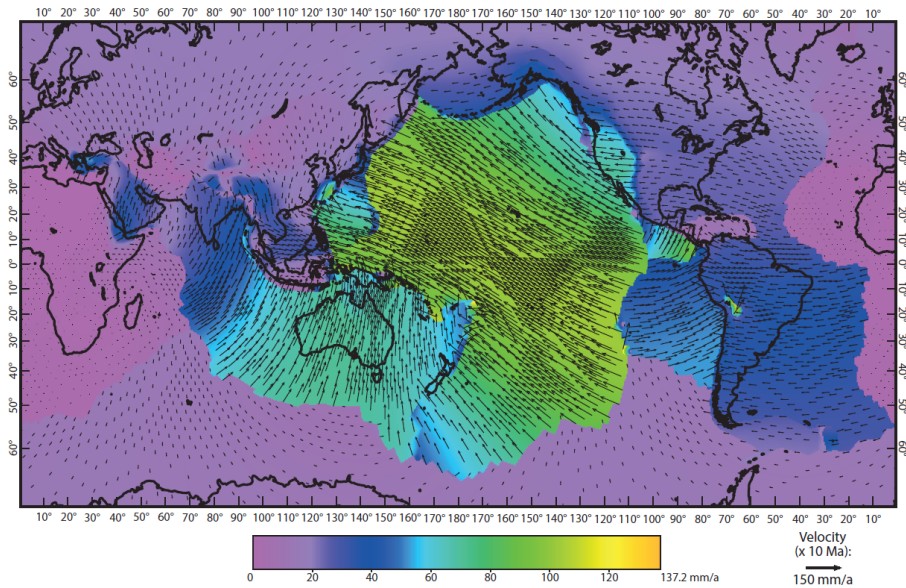

**Figure 6.** Surface horizontal velocity field of model 28.

As in example 1 we have plotted the horizontal velocities of the best model (model 28) in Fig. 6 for simple comparison with Fig. 4 and Fig. 10 of Bird et al. (2008). This model shows a further reduction in maximum horizontal velocity to approximately

137 mm $a^{-1}$ and, unlike in example 1, shows similar velocity magnitude results around Nepal to Bird et al. (2008).

By comparing our two examples we can see that, even though the best models are those with minimum fFric and tauMax, the addition of the fault slip rate data set causes the grid search (in this 2D fFric-tauMax space) to prefer models with a lower tauMax value.

## 5    Performance analyses

In this section we demonstrate two different performance tests used to check the performance of ShellSet. Firstly, we test the effect on both the total program and solution routine time that changing the number of MKL threads has. Secondly, we analyse the program performance when running multiple models in parallel. All performance results reported in this section are an average of 3 repeats of equal performance tests. Each section uses an equal model for each of its tests and this model is consistent over all sections.

### 5.1    MKL performance testing

We first test the effect that altering the number of threads available to the MKL routines has on both the routine itself and the total program time. We focus this analysis on the *dgbsv* routine as this is iteratively applied to the complete system of equations

representing the finite element grid ($16008 \times 16008$ in our global example), whereas the *dgesv* routine is used to solve a $3 \times 3$ system.

In order to analyse the effect of the MKL thread number we perform an experiment using a single model iterated twice on a single core (using a single MPI process), varying only the number of threads available to the MKL routines. Within an MKL routine the number of threads used includes the core working in serial, so a team of 64 MKL threads is the serial running core plus an extra 63 threads.

The experiment was performed on a single 64-core node of the HPC cluster at INGV Rome. We use a single node as MKL threads work in a shared memory environment and so cannot be divided across multiple nodes. Table 4 shows the time taken and calculated speedup (SU) for both the entire program and the *dgbsv* routine for the model when run with 1-64 MKL threads, while Fig. 7 plots these results.

**Table 4.** Performance tests performed using a 64-core node. All simulations were performed using a model on a single node, varying only the number of MKL threads. The time, in seconds, and speedup are reported to 2 decimal places.

| MKL threads | *dgbsv* time (s) | *dgbsv* SU | Prog. time (s) | Prog. SU |
|---|---|---|---|---|
| 1 | 120.19 | $\times$ | 190.97 | $\times$ |
| 2 | 66.93 | 1.80 | 135.51 | 1.41 |
| 4 | 39.14 | 3.07 | 108.74 | 1.76 |
| 8 | 25.87 | 4.65 | 96.46 | 1.98 |
| 16 | 21.82 | 5.51 | 91.60 | 2.09 |
| 32 | 26.41 | 4.55 | 96.38 | 1.98 |
| 64 | 44.54 | 2.70 | 114.31 | 1.67 |

Tests performed on a 4-socket system equipped with (4) Xeon Gold 5218 CPUs which are 16 core/32 thread chips.

Looking at Table 4 we can see that increasing the number of MKL threads does improve the performance over a single model, up to a point. Increasing the number of MKL threads from 1 to 16 would see a speedup of approximately 5.5, reducing the time spent inside the MKL routine from 120 s to 22 s and accordingly reducing the time of the entire program by approximately 100 s, which is not insignificant for a single model. Furthermore, it would accumulate to a significant time saving over a large number of models. Adding extra threads to the MKL routines beyond 16 began to degrade performance potentially due to the fact that the HPC node is comprised of 4 CPUs with 16 cores and so using more than 16 threads would mean performing calculations across CPU boundaries. This can degrade performance through requiring cross-socket (inter-CPU) memory accesses and thread barriers which can offer reduced performance relative to using a single CPU.

## 5.2 MPI performance testing

ShellSet causes no appreciable delay when individual components are compared to the three original programs as the underlying software is not changed. However, the inter-program links within ShellSet offer two opportunities for a performance

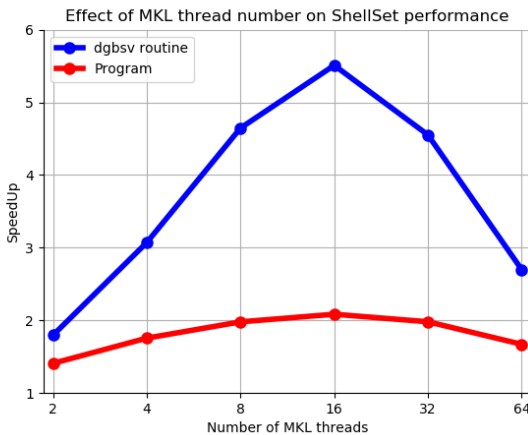

**Figure 7.** Speedup plot for the *dgbsv* routine and program using 2-64 MKL threads.

increase when compared to the original program. Firstly, the total simulation time of each model is reduced thanks to the
removal of the user interface during simulations. Secondly, the MPI framework made possible by these new links allows
numerous models to be run in parallel. We now provide analysis of these two enhancements.

### 5.2.1 Performance gain from removing user interface

Removing the need for a user interface between the formerly independent program units decreases the time required to complete
any model or set of models. The links, which would otherwise require a user to feed output from one program into the next,
are now performed automatically by ShellSet. Comparing the time taken by ShellSet to utilise these links to a user is a tricky
task as the time taken depends on the user and their abilities so it is not something which we will show, however we report the
following comparison from our experience.

Table 2 of Bird et al. (2008) shows 18 models in which the fFric and tauMax variables were varied. Each model required one
run of OrbData, 4-5 iterations (as stated in the text) of Shells and one run of OrbScore. With all file IO controlled manually by
the user these 18 models (72-90 Shells runs) took four days to one week when performed in a serial manor. We have performed
a realistic test where we alter both the number of MPI processes and MKL threads, testing the performance over 18 models on
a desktop machine.

Following the approach outlined in Bird et al. (2008) each timed model consists of: using OrbData to perform an update to
the finite element grid; four Shells iterations; a fifth Shells iteration (using updated boundary conditions); and a single OrbScore
run to score the final run. We use a model that we know completes fully, meaning it converges within the user defined MKL
iteration limit and passes all input variable criteria.

Table 5 reports the time taken for various MPI process and MKL thread pairings to perform 18 models. It shows that ShellSet
is able to perform 18 models with 5 Shells iterations in less than 90 minutes and as fast as 48 minutes. Although the results

shown in Table 5 were performed on more recent hardware and software, a large portion of the time taken for the 18 models of Bird et al. (2008) was lost to user interface and times where the user was not present - these are not issues with ShellSet.

**Table 5.** Performance test performed using an Intel Core i9-12900 CPU at 2.4GHz desktop with 64GB RAM, 16 physical cores (8 performance, 8 efficient) and 24 threads. The times are reported to the nearest second (mm:ss).

| Processes | 18 Models |
|:---:|:---:|
| 1 | 88:25 |
| 2 | 58:55 |
| 4 | 47:36 |
| 8 | 49:09 |
| 16 | 53:03 |

1 worker uses 6 MKL threads; 2 workers use 4 MKL threads; 4 workers use 2 MKL threads; 8 & 16 workers use 1 MKL thread.

### 5.2.2 Performance gain from parallel models

The use of an MPI framework provides the second opportunity for a performance improvement by allowing multiple models to be run in parallel. This can be tested in a more traditional fashion by comparing the time taken to complete a given number of models using a given number of processes. We also report the ratio between the serial and parallel times, known as the speedup.

For simplicity the tests were performed over a single search level of a 2D parameter space, varying the number of models and the number of worker processes. We do not report the results where the number of processes is greater than models as ShellSet contains a check on the number of models and worker processes to prevent an over-allocation of compute resources, in reality it would likely have a negative effect by unnecessarily tying up compute resources. For these tests we have fixed the number of MKL threads to 1 in order to maximise the number of cores available for MPI processes, we have also fixed all models to be equal (after the model generation procedure) in order to be able to more fairly compare timing results across all numbers of models, processes and both machines used. Each of the tests iterates the Shells solution procedures twice.

Table 6 reports the time in minutes and seconds for the test performed on a *typical* desktop computer, bracketed entries are the speedup compared to a single worker. Table 7 reports only the speedup for the test carried out on a 64-core HPC node. We calculate the speedup (SU) using:

$$SU = \frac{T_s}{T_p}$$

where $T_s$ is the time taken by a single worker and $T_p$ is the time taken by multiple workers.

The speedup values reported (bracketed) in Table 6 are around 2-4. In fact, the speedup results for 2 and 4 processes are very reasonable at approximately 1.9 and 3.3 respectively. Increasing the number of processes beyond this however yields very

**Table 6.** Performance test performed using an Intel Core i9-12900 CPU at 2.4GHz desktop with 64GB RAM, 16 physical cores (8 performance, 8 efficient) and 24 threads. Each time is reported to the nearest second, in mm:ss, and is an average of 3 tests. The speedup, shown in brackets, is calculated using seconds to 5 decimal places but reported to 2.

| Workers | Models | | | | | |
|---|---|---|---|---|---|---|
| | 2 | 4 | 8 | 16 | 32 | 64 |
| 1 | 6:25 | 12:49 | 25:41 | 51:11 | 102:19 | 205:13 |
| 2 | 3:18 (1.95) | 6:42 (1.91) | 13:40 (1.88) | 27:35 (1.86) | 55:15 (1.85) | 110:07 (1.86) |
| 4 | × | 3:57 (3.25) | 7:47 (3.30) | 15:54 (3.22) | 32:54 (3.11) | 66:04 (3.11) |
| 8 | × | × | 6:49 (3.77) | 13:06 (3.91) | 26:20 (3.89) | 52:02 (3.94) |
| 16 | × | × | × | 13:29 (3.80) | 26:48 (3.82) | 52:27 (3.91) |

All simulations were performed with the number of threads used within MKL routines fixed at 1.

little in extra speedup and eventually a decrease in performance, see Fig. 8 for a plot of the speedup. This could be because this computer is relatively small (with 8 performance and 8 efficient cores) and increasing the number of processes starts to strain the system. By increasing the number of MPI processes performing models in parallel without increasing the total machine size we are increasing the load on the memory bandwidth. This will eventually cause a transition from a compute-bound to a memory-bound situation which would seem to occur between 4 and 8 MPI processes, which would then explain the levelling of performance after 8 processes. It is also the case that this test was performed in a WSL2 Linux environment hosted on Windows and therefore we cannot be certain about the true availability of the cores since there will be some background OS work related to Windows, not least the running of the WSL2 environment.

Since the general user does not usually worry about speedup results and instead focuses mostly on time, and despite poor speedup results for a larger number of processes, the results using up to 4 and even 8 processes are very interesting. The small increase in performance between 4 and 8 processes is worth a considerable time saving when performing 64 models with roughly 14 minutes saved, and over 6 minutes saved for 32 models.

The speedup values reported in Table 7, which shows results obtained on a single HPC node, range from 2 to almost 30. The speedup results for 2, 4, 8 and 16 processes are very good at approximately 2, 3.8, 7.2 and 13.6 respectively, remaining close to the ideal values. As in Table 6, increasing the number of processes beyond this started to see less increase in performance but not to the same extent. Even at 32 processes the speedup is reasonable at approximately 22. We only see poor performance at 64 cores - when the node is full. Figure 9 shows a plot of the speedup values. Interestingly, we can see that unlike in Fig. 8 where the three plots have the same behaviour, in Fig. 9 the behaviour is equal until the node is full where we see quite a large difference in performance between the three plots. This is possibly a symptom of the number of models performed within the test, where running more models hides some poor performance as a result of some models taking longer than others to complete, although this would likely be borne out in other results as well. Another possibility is a more general unpredictability when a compute node is full.

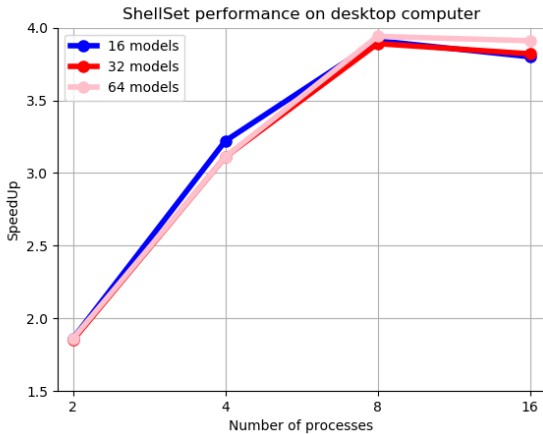

**Figure 8.** Speedup plot for 2, 4, 8 & 16 workers performing 16, 32 & 64 models, on desktop machine. Values are given (in brackets) in Table 6.

**Table 7.** Speedup results from performance tests performed using a 64-core node. The node is a 4-socket system equipped with (4) Xeon Gold 5218 CPUs which are 16 core/32 thread chips. The speedup is calculated using seconds to 5 decimal places but reported to 2.

| Workers | Models | | | | | | | |
|---|---|---|---|---|---|---|---|---|
| | 2 | 4 | 8 | 16 | 32 | 64 | 128 | 256 |
| 2 | 2.00 | 1.99 | 2.00 | 2.00 | 1.98 | 2.01 | 2.01 | 2.01 |
| 4 | × | 3.84 | 3.83 | 3.84 | 3.82 | 3.86 | 3.89 | 3.92 |
| 8 | × | × | 7.02 | 7.10 | 7.12 | 7.19 | 7.27 | 7.43 |
| 16 | × | × | × | 13.44 | 13.65 | 13.81 | 13.75 | 13.77 |
| 32 | × | × | × | × | 22.09 | 22.34 | 22.58 | 22.51 |
| 64 | × | × | × | × | × | 18.63 | 23.88 | 27.69 |

All simulations were performed with the number of threads used within MKL routines fixed at 1.

We can see in these results the benefit of a larger, and dedicated, compute system over a smaller hosted environment with the speedup results of Table 7 remaining much closer to the ideal value for longer. One possible reason for the slowing of performance increase, seen between 16 and 32 MPI processes, is that, as noted in relation to the previous MPI performance example, we are increasing the load on the memory bandwidth and possibly transitioning from a compute-bound to a memory-bound situation. A reason for, in comparison to the first MPI performance test, this levelling of performance occurring at a higher number of MPI processes could be that the HPC system has a higher memory-bandwidth. It is also true that there will be fewer interfering background processes on a dedicated compute system compared to WSL2 hosted in Windows OS.

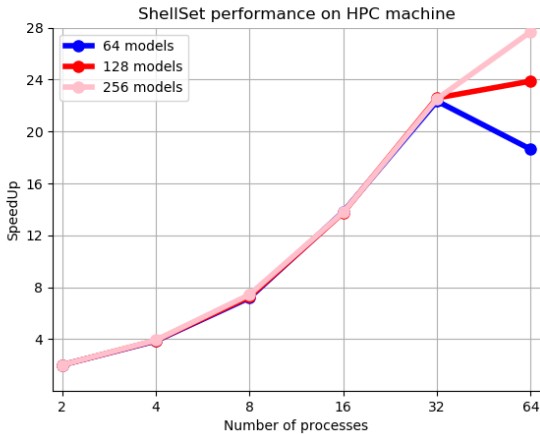

**Figure 9.** Speedup plot for 2, 4, 8, 16, 32 & 64 workers performing 64, 128 & 256 models, on 64-core node of HPC machine. Values are given in Table 7.

The two tests using two different types of Linux environment on two different machines prove that there is a considerable
time saving to be made by running models in parallel. Our tests have shown results which are 4 times faster on a *typical* desktop machine and almost 30 times faster using a dedicated compute node.

In general, searching algorithms which contract through levels such as the grid search could suffer performance loss at the interface between the end of one level and the beginning of the next. Indeed, the grid search employed in ShellSet would suffer from such a degradation, however the examples shown use a single level to generate the number of models required and so this
would not be present in these performance results.

Taken together, the results of the MKL and MPI performance experiments suggest that, at least for our example global model, it is more important to run many models in parallel rather than fewer highly parallel models. Since the speedup performance shown in Table 4 quickly falls away from the ideal value, whereas the speedup results shown in Table 6 and Table 7 remain close to the ideal value, we can see that the most important aspect of parallelism for ShellSet is a many models approach.
Trivially, when running a relatively small number of models on a large enough machine the most sensible approach is likely to be a combination of the two parallel schemes by performing muli-threaded MKL routine calls in parallel. We note that the best approach will depend on the underlying model size, its complexity, and the size and technical characteristics of the compute machine, all of which will be personal to the user.

## 6   Conclusions and future work

This article has outlined a new MPI parallel dynamic neotectonic modelling software, ShellSet, which has been designed for use by the wider geodynamic modelling community. The simplified *hands-off* nature of ShellSet reduces user-program

interactions to a minimum, while the addition of a GUI further simplifies those remaining interactions. These improvements open ShellSet to a less experienced user, while also reducing setup errors for all users.

We have shown in Sect. 4 two examples of ShellSet's abilities within the target field of study by first improving on an existing global model developed in Bird et al. (2008), then further improving on that with the addition of a new data set on fault slip rates. Both examples were completed in a fraction of the time taken to generate the existing global model of Bird et al. (2008).

In upcoming work ShellSet will be used to optimise the rheology of the lithosphere in the Apennine mountain region for better dynamical simulations of neotectonics and associated seismicity. We also plan to use ShellSet in work forming predictions of seismic hazard within the Central Mediterranean region.

ShellSet was developed and tested on a computer with widely available capabilities. We have demonstrated that ShellSet achieves an improved performance, relative to the original software, and speedup when running multiple models in parallel on both *typical* and larger HPC machines, see Sect. 5. Despite the improved performance, in Sect. 5 we have noted potential reasons for the levelling off in performance when running multiple models in parallel, one of which potentially being that we are crossing from a compute-bound to a memory-bound situation. The grid search algorithm could also cause performance loss at the transition between contracting levels, however our tests did not analyse this possibility. An analysis of parameter space sampling methods, as done in Reuber (2021), could yield a method which would allow ShellSet to better utilise its parallelism, particularly on high-performance computing (HPC) machines. One candidate method is the NAplus algorithm, see Baumann et al. (2014), which is shown to scale well and is built on the well-known Neighbourhood algorithm introduced in Sambridge (1999). Another is a random walk sampling performed using multiple parallel and independent random walks. While these sampling methods would not solve the issue of performance loss when transitioning into a memory-bound situation, more efficient HPC machine use would enable searching the parameter space to a finer level, faster results, and the improved use of larger HPC structures would allow use of much larger (in terms of memory) models.

*Code and data availability.* The exact version of ShellSet used to produce the results shown in this paper is archived on Zenodo, see https://zenodo.org/records/7986808, May et al. (2023), along with all input files, scripts to compile and run ShellSet, the GUI and the scatter plotting routine. The most current version of ShellSet is always available from the project website: https://github.com/JonBMay/ShellSet under the GPL-3.0 license. At time of publication this version is identical to the Zenodo archive except for an updated Makefile, which now points to a general installation location for MKL; User Guide updates which reflect this update and provide information on what to change in the Makefile for other MKL installation locations; and minor README updates.

## Appendix A: Fault slip rate (FSR) data set

Here we outline the process followed to create a usable fault slip rate data set, as used in example 2 of Sect. 4.

In Styron and Pagani (2020) the authors report a global active fault database of approximately 13,500 faults collected through the combination of regional data sets. The authors note that approximately 77% of the faults have slip rate information, this provided us with an estimated initial 10395 faults for potential use in our work.

Due to the requirements of our program the database needed some cleaning. Of the approximately 13,500 initial fault traces,
around 9,500 faults either have no offset rate or lack upper and lower limits which identify a rate based on a dated offset feature
(unbounded rates are model rates, which we do not consider to be data.) A further 684 lack rake info and 1022 fault traces lack
necessary dip information. After discounting these fault traces, we were left with 2487 traces which might be used in OrbScore
if each aligns with a fault element of the current finite element grid.

Those fault traces remaining are mostly located in one of four areas: the Mediterranean/Tethyan orogenic belt, well covered
from Portugal to Pakistan; Japan; New Zealand; and California. Many of these almost 2,500 fault traces (e.g. those in Japan)
did not correspond in position (and orientation) to any of the fault elements of the existing finite element grid as used in Bird
et al. (2008). Therefore, they were also removed from our data set.

Remaining fault traces were subjected to the following two comparison filters with respect to the finite element grid of 2008:
firstly, the overall azimuth (endpoint-to-endpoint) of the fault trace must be within a $\pm$ 15 degrees tolerance of the grid fault
element azimuth; secondly, the shortest distance from the fault trace to the grid fault element must be less than one-eighth of
the length of that fault element.

After applying these conditions to the fault traces, we were left with 931 associations. Accounting for the fault traces of the
database which match with multiple fault elements left us with 572 faults.

The fault elements of the finite element grid file that have multiple associations within the new data set needed to be
controlled to avoid over-weighting of those elements with more associations. This was done by summing the product of
"adjacent" fault length (limited to the length of the fault-element) with the offset rate for all the associated traces, then dividing
the sum by the length of the fault element to get the aggregated offset rate that the fault element should match.

**Appendix B: Tables related to demonstrated real world examples**

**Table B1.** Example 1 grid search results, excluding the first level which is shown in Table 2.

| Global Model Number | fFric | tauMax (N m$^{-1}$) | SSR (mm a$^{-1}$) | GV (mm a$^{-1}$) | SD (deg.) | SA (deg.) | GM |
|---|---|---|---|---|---|---|---|
| 10 | 0.06806 | $1.50 \times 10^{12}$ | 7.14 | 12.04 | 31.28 | 27.52 | 16.50 |
| 11 | 0.15417 | $1.50 \times 10^{12}$ | 9.43 | 12.38 | 32.09 | 25.69 | 17.61 |
| 12 | 0.24028 | $1.50 \times 10^{12}$ | 11.75 | 13.45 | 32.80 | 26.23 | 19.20 |
| 13 | 0.06806 | $2.50 \times 10^{12}$ | 8.06 | 13.19 | 30.49 | 26.12 | 17.06 |
| 14 | 0.15417 | $2.50 \times 10^{12}$ | 9.97 | 13.02 | 31.10 | 24.93 | 17.81 |
| 15 | 0.24028 | $2.50 \times 10^{12}$ | 12.18 | 14.17 | 32.00 | 24.83 | 19.24 |
| 16 | 0.06806 | $3.50 \times 10^{12}$ | 8.47 | 14.02 | 30.24 | 25.26 | 17.35 |
| 17 | 0.15417 | $3.50 \times 10^{12}$ | 10.79 | 14.80 | 30.25 | 24.13 | 18.48 |
| 18 | 0.24028 | $3.50 \times 10^{12}$ | 13.09 | 16.28 | 31.59 | 24.09 | 20.07 |
| 19 | 0.06806 | $4.50 \times 10^{12}$ | 10.11 | 19.12 | 30.36 | 24.49 | 19.47 |
| 20 | 0.15417 | $4.50 \times 10^{12}$ | 12.31 | 19.80 | 30.61 | 23.39 | 20.44 |
| 21 | 0.24028 | $4.50 \times 10^{12}$ | 14.52 | 20.56 | 31.61 | 23.25 | 21.64 |
| 22 | 0.06806 | $5.50 \times 10^{12}$ | 11.35 | 21.05 | 30.87 | 23.89 | 20.49 |
| 23 | 0.15417 | $5.50 \times 10^{12}$ | 13.65 | 21.85 | 31.04 | 23.04 | 21.49 |
| 24 | 0.24028 | $5.50 \times 10^{12}$ | 16.02 | 22.64 | 31.57 | 22.67 | 22.57 |
| 25 | 0.06806 | $6.50 \times 10^{12}$ | 11.84 | 21.57 | 31.41 | 23.35 | 20.80 |
| 26 | 0.15417 | $6.50 \times 10^{12}$ | 14.39 | 22.75 | 31.40 | 22.90 | 22.03 |
| 27 | 0.24028 | $6.50 \times 10^{12}$ | 17.19 | 23.80 | 32.12 | 22.34 | 23.27 |
| 28 | 0.03935 | $1.17 \times 10^{12}$ | 6.57 | 12.27 | 32.00 | 27.65 | 16.34 |
| 29 | 0.06806 | $1.17 \times 10^{12}$ | 7.03 | 11.70 | 31.57 | 27.10 | 16.29 |
| 30 | 0.09676 | $1.17 \times 10^{12}$ | 7.59 | 11.64 | 31.49 | 26.94 | 16.55 |
| 31 | 0.03935 | $1.50 \times 10^{12}$ | 6.69 | 12.68 | 31.73 | 27.50 | 16.49 |
| 32 | 0.06806 | $1.50 \times 10^{12}$ | 7.14 | 12.04 | 31.28 | 27.52 | 16.50 |
| 33 | 0.09676 | $1.50 \times 10^{12}$ | 7.74 | 11.96 | 31.03 | 26.35 | 16.58 |
| 34 | 0.03935 | $1.83 \times 10^{12}$ | 6.89 | 13.04 | 31.46 | 27.23 | 16.65 |
| 35 | 0.06806 | $1.83 \times 10^{12}$ | 7.31 | 12.27 | 30.99 | 27.01 | 16.55 |
| 36 | 0.09676 | $1.83 \times 10^{12}$ | 7.86 | 12.09 | 30.87 | 25.66 | 16.56 |
| 37 | 0.03935 | $2.17 \times 10^{12}$ | 7.17 | 13.43 | 31.21 | 27.00 | 16.88 |
| 38 | 0.06806 | $2.17 \times 10^{12}$ | 7.56 | 12.64 | 30.65 | 26.58 | 16.70 |
| 39 | 0.09676 | $2.17 \times 10^{12}$ | 8.10 | 12.44 | 30.48 | 25.29 | 16.69 |
| 40 | 0.03935 | $2.50 \times 10^{12}$ | 7.56 | 13.95 | 31.04 | 26.82 | 17.22 |
| 41 | 0.06806 | $2.50 \times 10^{12}$ | 8.06 | 13.19 | 30.49 | 26.12 | 17.06 |
| 42 | 0.09676 | $2.50 \times 10^{12}$ | 8.63 | 12.96 | 30.33 | 24.88 | 17.05 |
| 43 | 0.03935 | $2.83 \times 10^{12}$ | 7.60 | 14.23 | 31.07 | 27.35 | 17.41 |
| 44 | 0.06806 | $2.83 \times 10^{12}$ | 8.21 | 13.48 | 30.40 | 25.65 | 17.14 |
| 45 | 0.09676 | $2.83 \times 10^{12}$ | 8.95 | 13.41 | 30.19 | 24.58 | 17.28 |

Misfit types listed in Table 1

**Table B2.** Example 2 grid search results, excluding the first level which is shown in Table 3.

| Model | fFric | tauMax (N m$^{-1}$) | SSR (mm a$^{-1}$) | GV (mm a$^{-1}$) | SD (deg.) | SA (deg.) | FSR (mm a$^{-1}$) | GM |
|---|---|---|---|---|---|---|---|---|
| 10 | 0.06806 | $1.50 \times 10^{12}$ | 7.14 | 12.04 | 31.28 | 27.52 | 4.80 | 12.89 |
| 11 | 0.15417 | $1.50 \times 10^{12}$ | 9.43 | 12.38 | 32.09 | 25.69 | 3.63 | 12.85 |
| 12 | 0.24028 | $1.50 \times 10^{12}$ | 11.75 | 13.45 | 32.80 | 26.23 | 3.20 | 13.42 |
| 13 | 0.06806 | $2.50 \times 10^{12}$ | 8.06 | 13.19 | 30.49 | 26.12 | 4.98 | 13.33 |
| 14 | 0.15417 | $2.50 \times 10^{12}$ | 9.97 | 13.02 | 31.10 | 24.93 | 3.84 | 13.10 |
| 15 | 0.24028 | $2.50 \times 10^{12}$ | 12.18 | 14.17 | 32.00 | 24.83 | 3.33 | 13.55 |
| 16 | 0.06806 | $3.50 \times 10^{12}$ | 8.47 | 14.02 | 30.24 | 25.26 | 5.68 | 13.88 |
| 17 | 0.15417 | $3.50 \times 10^{12}$ | 10.79 | 14.80 | 30.25 | 24.13 | 4.40 | 13.87 |
| 18 | 0.24028 | $3.50 \times 10^{12}$ | 13.09 | 16.28 | 31.59 | 24.09 | 3.62 | 14.24 |
| 19 | 0.32639 | $1.50 \times 10^{12}$ | 14.61 | 14.66 | 33.78 | 26.32 | 3.16 | 14.32 |
| 20 | 0.41250 | $1.50 \times 10^{12}$ | 18.22 | 15.84 | 34.67 | 26.27 | 3.27 | 15.37 |
| 21 | 0.49861 | $1.50 \times 10^{12}$ | 21.68 | 17.40 | 35.14 | 26.01 | 3.25 | 16.21 |
| 22 | 0.32639 | $2.50 \times 10^{12}$ | 15.25 | 16.03 | 33.10 | 24.99 | 3.11 | 14.45 |
| 23 | 0.41250 | $2.50 \times 10^{12}$ | 18.84 | 17.43 | 33.82 | 24.90 | 3.15 | 15.41 |
| 24 | 0.49861 | $2.50 \times 10^{12}$ | 22.18 | 18.94 | 34.28 | 24.85 | 3.17 | 16.25 |
| 25 | 0.32639 | $3.50 \times 10^{12}$ | 16.32 | 18.42 | 32.50 | 23.78 | 3.32 | 15.05 |
| 26 | 0.41250 | $3.50 \times 10^{12}$ | 19.90 | 19.88 | 33.14 | 23.59 | 3.10 | 15.72 |
| 27 | 0.49861 | $3.50 \times 10^{12}$ | 23.11 | 21.20 | 33.84 | 23.79 | 3.08 | 16.48 |
| 28 | 0.12546 | $1.17 \times 10^{12}$ | 8.46 | 11.85 | 32.06 | 25.78 | 3.93 | 12.66 |
| 29 | 0.15417 | $1.17 \times 10^{12}$ | 9.28 | 12.16 | 32.36 | 25.93 | 3.60 | 12.78 |
| 30 | 0.18287 | $1.17 \times 10^{12}$ | 10.03 | 12.51 | 32.58 | 26.09 | 3.38 | 12.93 |
| 31 | 0.12546 | $1.50 \times 10^{12}$ | 8.63 | 12.12 | 31.74 | 25.46 | 3.97 | 12.74 |
| 32 | 0.15417 | $1.50 \times 10^{12}$ | 9.43 | 12.38 | 32.09 | 25.69 | 3.63 | 12.85 |
| 33 | 0.18287 | $1.50 \times 10^{12}$ | 10.17 | 12.71 | 32.28 | 25.84 | 3.41 | 12.97 |
| 34 | 0.12546 | $1.83 \times 10^{12}$ | 8.84 | 12.36 | 31.29 | 24.99 | 4.02 | 12.80 |
| 35 | 0.15417 | $1.83 \times 10^{12}$ | 9.57 | 12.59 | 31.76 | 25.35 | 3.69 | 12.90 |
| 36 | 0.18287 | $1.83 \times 10^{12}$ | 10.35 | 12.97 | 31.93 | 25.53 | 3.45 | 13.04 |
| 37 | 0.03935 | $1.17 \times 10^{12}$ | 6.57 | 12.27 | 32.00 | 27.65 | 5.31 | 13.05 |
| 38 | 0.06806 | $1.17 \times 10^{12}$ | 7.03 | 11.70 | 31.57 | 27.10 | 4.78 | 12.74 |
| 39 | 0.09676 | $1.17 \times 10^{12}$ | 7.59 | 11.64 | 31.49 | 26.94 | 4.35 | 12.67 |
| 40 | 0.03935 | $1.50 \times 10^{12}$ | 6.69 | 12.68 | 31.73 | 27.50 | 5.32 | 13.16 |
| 41 | 0.06806 | $1.50 \times 10^{12}$ | 7.14 | 12.04 | 31.28 | 27.52 | 4.80 | 12.89 |
| 42 | 0.09676 | $1.50 \times 10^{12}$ | 7.74 | 11.96 | 31.03 | 26.35 | 4.39 | 12.72 |
| 43 | 0.03935 | $1.83 \times 10^{12}$ | 6.89 | 13.04 | 31.46 | 27.23 | 5.34 | 13.26 |
| 44 | 0.06806 | $1.83 \times 10^{12}$ | 7.31 | 12.27 | 30.99 | 27.01 | 4.83 | 12.94 |
| 45 | 0.09676 | $1.83 \times 10^{12}$ | 7.86 | 12.09 | 30.87 | 25.66 | 4.44 | 12.73 |

Misfit types listed in Table 1

*Author contributions.* JBM wrote the ShellSet code and additional Python routines, designed and ran the examples and performance tests, wrote the user guide, and wrote the first draft of the article after which all authors contributed. PB and MMCC supervised the correct combination of software into the final program. PB generated the new fault slip rate data set, and generated map figures. MMCC devised the work.

*Competing interests.* The authors declare that they do not have any competing interests.

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
