# Peer review of "ShellSet v1.1.0 - Parallel Dynamic Neotectonic Modelling: A case study using Earth5-049"

_EGUsphere, 2023_

## Referee Comment (RC1)

[referee-annotated manuscript omitted]

---

## Author Response (AR1)

**Referee #1 (10/11/23)**

• The ShellSet program, following its predecessor SHELLS, is able to model lateral variations of the lithospheric strength, in terms of laterally-varying crustal and lithospheric thickness, elevation and heat-flow. However, to my understanding, crustal and mantle density values, or fault parameters (i.e. fault friction) should be kept constant in the whole numerical domain, meaning that, for instance, for each model, fault elements should have the same friction coefficient in the whole domain. This can be an issue in large continental scale areas. Is there a way, in this new program, to laterally varying these parameters in the sense that users can, for instance, model each fault with customized friction parameters?

ShellSet does not assume laterally-constant density in either the crust or the mantle-lithosphere. Firstly, density at every point in the lithosphere is affected by ambient temperature, which depends on depth, local heat-flow, radioactive heat production in that layer, and any locally non-steady-state component of the geotherm (where heat-flow and non-steady components are computed by OrbData, the heat-production of each layer is an input parameter, and depth is relative to known topography). Secondly, the program OrbData also adds a "lithospheric density anomaly of chemical origin" at each node, which is adjusted to achieve isostasy with other nodes, given the crust and mantle-lithosphere thicknesses that have already been determined by OrbData. These chemical density anomalies are limited to small amplitudes (-50 to +50 kg/m$\hat{3}$) to avoid unreasonable results over trenches and mantle plumes, which are allowed to be non-isostatic instead. As for fault friction, ShellSet also allows different friction on different faults, if the Lithospheric Rheology #n (LRn) input option is used. Potentially, every fault element could be assigned a different friction. The qualification to this flexibility is that ShellSet will only vary and optimize Lithospheric Rheology 0 (the baseline, or default rheology), so in a model with different frictions on different faults, only one set of faults (or the surrounding domain) could be optimized in any run.

**We have noted the variable Lithospheric Rheology option of Shells (and ShellSet) in the text. An explanation of this is already present inside the user guide.**

• Rows 35-37, and 150: Is ShellSet able run on Mac OS? and which Python version is needed?

While we are quite confident that if an environment is correctly set up for ShellSet (guest Linux OS, e.g., using Oracle VM Virtual Box, required compilers and libraries etc) it will function on any host OS which can provide these (including Mac OS). However, we do not have access to any Mac machine to test this and therefore we cannot state for certain whether it would function or not. We agree that functionality on a Mac OS system would be a useful addition to ShellSet and will actively work to secure access to a suitable machine to test this for future updates to ShellSet. In the meantime, we remain contactable to aid users should issues arise on any OS. The Python version used is Python 3.8, the routine "ShellSetScatter" uses the libraries: numpy, re, os, matplotlib and mpl_toolkits.mplot3d, while "ShellSetGUI" uses: tkinter and os. We will note the Python version within the article text.

**Information related to version number for Python routines has been added to the GitHub page, and the article.**

• Rows 60-61: what kind of test is performed?

We use "test" to mean a set of models, each model being a fixed set of parameters. So, for example, a grid search is a single test of N models where we search a parameter space for the optimal set of values. We will make this clearer in the text.

**We have added a statement to the introduction stating these definitions we are using for *test* and *model* within the article.**

• Section 3: the inclusion of a flow diagram (like the one in Figure 1 of the UserGuide) to help the readers/users understand how to use it in an optimal way, would be helpful.

We will add a simple flow diagram to show how the elements of the ShellSet program link together.

**A flowchart showing how the original programs are connected, as well as the scope of the influence of the GUI & file used by the scatter plotter routine, has been added to section 3.**

• Rows 120-122: the mentioned "separate file" can be modified by the user? Which are "the most general conditions on the variable" and which are those can be modified by the users? Please, explain this better.

The separate file (as all source code) is modifiable by the user. The current set conditions are: the fault friction must be less

than the continuum friction; the crustal mean density must be less than the mantle mean density. We recommend that any other conditions which users may require be added to this file to 1) avoid requiring any modifications at other source code locations in order to correctly apply their checks & 2) to maintain a consistent location for this program update for simplified further modifications/debugging etc. We have placed no limit on the number of conditions or how the parameters within each condition interact.

**We have added information related to the conditions tested for by ShellSet within this new source file.**

**Referee #1 attachment**

- Add info on the code and data availability in the abstract

**The zenodo link was added to the abstract.**

- Line 8: In the first example, we ... [..], in the second example ...

**Done.**

- Line 24: many: please provide examples of existing codes and their references

**We have provided references to existing software.**

- Line 32: (created by OrbWin): OrbData5 ...

**Updated.**

- Line 33: "both": there are more than two programs cited here. Please, review English

**Both was referring to OrbScore & OrbData, we understand the confusion due to the text and have re-written for clarity.**

- Line 37: Only link for Shells is specified here. But the object of the work is ShellSet. Add link where ShellSet is available, also specifying that a tutorial is available there (if so).

**We have added a reference to the Zenodo repository released with this article.**

- Line 64: add references

**We have added references to software using 3D grids.**

- Line 74: dgbsv & dgesv in cursive

**Done.**

- Line 95: to individually control each one in separate instances (if I understand correctly)

**Correct. We have clarified this in the text.**

- Line 98 & 99: dgbsv & dgesv in cursive

**Done.**

• Line 98: levaraging

**We disagree that this is an error in the text so have made no change.**

• Line 102: MPI threads. From here ...

**This has been rewritten for clarity while also fulfilling a request from the 2nd reviewer to expand section 3.**

• Line 123: This is the main output from the program. I consider it should be more enphasized and put at the beginning of the section

**Since section 3 has been rewritten at the request of reviewer 2 the text referred to here is now the first improvement noted in the user interface section.**

• Line 131-133: 1) an input file, 2) an input, 3) a selection

**Done.**

• Line 138: please, rephrase

**We have added some detail & an example.**

• Line 142: Is it an extra package of ShellSet?

**It is part of the program as standard - this is now made clear in the text.**

• Line 150: please, rephrase

**Done.**

• Line 156: have added → add

**Done.**

• Line 169: was → has been

**This text was rewritten at the request of reviewer 2 and no longer exists in this form.**

• Line 182: Furthermore → Secondly (if there are two steps)

**This text was updated and moved by request of reviewer 2.**

• Line 184-186: Though there have been .. [..]..will affect the results, there have been numerous updates ...

**We have updated this text.**

• Line 186: Furthermore, Shellset is compiled ..

**Updated.**

• Line 190: report → reported

**Done.**

• Table 1 & 2 caption: note (Earth5-049)

**Updated.**

• Table 1 & 2: I would put in some evidence (bold type) the best model

**We have updated the best model row as bold text in both tables.**

• Table 1 & 2: Please, specify the meanings of each initials/acronyms

**Scoring types have been clarified in the table captions.**

• Line 223: This means that, in these areas, the new model shows lower velocities than in Bird et al., (2008), . .Is it what you mean? Please, explain it better

**After some consideration we realised that the horizontal velocity figures in the appendix where not the easiest to understand for a general reader and so we have updated them. Instead of showing images of differences between 2 horizontal velocity fields which are not easily interpreted we have included images which show the horizontal velocity field of the best model for each example. These figures are more immediately understandable while a comparison is easily made with the original results by comparing them with figure 10 of "Stresses that drive the plates from below: Definitions, computational path, model optimisation, and error analysis" (Bird et al. (2008)). These new figures have also been moved into the main body of the manuscript with some simple analysis, while the tables remain in the appendix.**

• Line 243: Could you please spend some more words to illustrate the figure?

**We have added more detail comparing the results of examples 1 & 2.**

• Line 246: However, the linkage ..

**This text has been updated.**

• Line 247: is lower thanks to the removal of the user interface.

**Updated.**

• Line 253: his? - or do you mean "users"?

**This was confusing as there was no "his" here. We are referring to the fact that the links between the independent programs are manually controlled by a user, however within ShellSet these links are handled automatically. We do not directly test the performance improvement from this update as the performance of the independent programs will depend strongly on the user and their programming skills with regards to correctly feeding the required data between 1 program and the next. We do provide an example of the increased performance based on experience.**

- Line 270-271: Please, rephrase

**Since the performance tests were altered at the request of the 2nd reviewer this statement has been removed.**

- Table 3: Please, explain table 3 values & bracketed values better in the caption.

**We have clarified the table data within the table caption.**

- Line 306: extra (

**This was not an extra (, however we have reworded the text.**

- Line 332: please, rephrase

**This has been updated, and moved into section 3 as suggested by reviewer 2.**

**Referee #2: Rene Gassmoeller (25/01/2024)**

• Line 34: The authors claim that ShellSet is entirely based on open source software, with a reference to a personal website of one of the authors. Unfortunately this is not sufficient to fulfil the definition of open source software as maintained by the Open Source Initiative (https://opensource.org/osd/). In particular the linked software does not include any license information (after a somewhat exhaustive search). This means that while the author clearly intends to make the software freely available, legally it is not spelled out what users are allowed to do (technically, the strictest copyright applies and users are not allowed to do any modification or distribute the software). This is not a problem for ShellSet itself though, since ShellSet is clearly licensed under GPL 3. I would suggest to either modify this statement to say "freely-distributed" dependencies, or to include an open source license on the linked website to make it clear to users what license this software is distributed under. I am aware that much of this software was written before clear definitions for open source software existed.

Due in part to this comment the author of the linked software is now in the process of updating the licensing of his works, beginning with the programs used in ShellSet. We will update the wording of the article to state that these components are "freely-distributed" but reserve the option to alter again if the licensing process is finalised before the final publication.

**We have changed the text to state that the component programs are freely-available.**

• Line 35: ShellSet requires Intel compiler, MPI, and MKL, but only provides a makefile with hard-coded include paths to Intel MKL. This approach is prone to problems on user systems since Intel MKL may be installed anywhere, in particular if the user is not an administrator of their system. A more portable solution would be to use cmake or autoconf as an operating system independent configuration system that allows to automatically find the location of Intel MKL. At least the authors should modify the Makefile so that it points to an include directory that also includes the Intel MPI headers and document how to change this directory. On my system mpif.h could not be found after adjusting the include paths, because the Makefile links to the MKL include directory, not the general Intel OneAPI include directory. I had to modify -I"/opt/intel/oneapi/mkl/2021.3.0/include" to -I/opt/intel/oneapi/2024.0/include, which is an unnecessary hurdle for new or inexperienced users. The (necessary) inclusion of version numbers in this path is another obstacle that results from using hand-written Makefiles.

An update will be made to the Makefile available in the package on GitHub which alters the include paths to use MKLROOT which is automatically defined during the installation of Intel MKL. The GitHub page is advertised on the Zenodo package page linked to the article. We will also add information within the user guide related to the issue noted and how/where to change the Makefile as exampled within the comment. Considering our plans for the software, as noted in the article and a later comment, in future we will alter the installation process to something such as cmake to simplify the installation process.

**The Makefile and user guide have been updated in the GitHub repository, which is advertised on the ShellSet Zenodo page.**

• Section 2.3 The description of OrbScore in the manuscript is insufficient. It is not laid out how exactly OrbScore compares to existing datasets and computes the scores of model results and the only reference to a description of OrbScore is given as a link to a personal website that cannot give a guarantee for future availability. At least the short text that is already available on the website should be included in the manuscript. In addition even on the linked website it is not spelled out "how" the score is computed (e.g. is it an RMS difference, or some other error norm? Is it the same norm for all criteria? How are the different criteria scaled against each other for the combined score?). I understand that some of the other tools have been published elsewhere, but there is no reference given to the original publication of OrbScore. **Later edit:** After continuing to read the manuscript I found the necessary reference in lines 181-182, which refer to the original description in Bird et al 2008. I think at least the paragraph that describes the grading procedure in line 179-183 should be generalized and moved into section 2.3 to explain OrbScore to the reader. My preference would be to also list all the available datasets (this gives the chance to explain any new datasets since the Bird et al. 2008 paper), the ways the individual scores are computed, and the procedure (and reasoning) for forming the final score as geometric mean. Lines 179-183 could then be shortened to say that OrbScore was run with the specific subset of datasets as in Bird et al. 2008.

We will add further details on OrbScore into section 2.3, including details on the scoring datasets provided in the package. For some clarity, OrbScore computes and reports a number of alternative metrics, but (based on years of experience) its author has selected one of these (for each kind of dataset) which is recommended* for model-ranking, and therefore that is the one reported to ShellSet. These choices can be changed within the OrbScore source code, by modifying a single line in each case. *For example, RMS misfit is only preferred (because of its relation to likelihood) if the errors in the scoring dataset have Gaussian distributions. But for many scoring datasets there are non-Gaussian "outliers" so that another metric, such as mean (absolute value) of misfit is preferred.

**We have expanded the section detailing OrbScore, including information regarding the scoring datasets, how each score is calculated by the program, and the units for each score.**

- Line 100: This sentence is somewhat confusing. I think what you are saying is that "by default" Intel MKL will automatically choose the maximum number of threads (e.g. as many as there are cores), but since you also run multiple MPI ranks in parallel it is better to let ShellSet choose the number of threads manually (e.g. as number of available cores divided by number of running MPI ranks). Please clarify the first part of the sentence.

The understanding within the comment is correct, however we agree that there could be some clarity in the wording we have used.

**We have clarified the information around the default behaviour of the MKL thread number selection and the ShellSet control to prevent an over-request.**

- Line 102+103: usually we speak of MPI processes (or ranks if you refer to the specific number), not threads. Threads are a different parallelization model, so MPI threads does not usually make sense (I think you refer to MPI processes in these two lines).

We will alter each "threads" reference to "processes" where necessary.

**We have changed "threads" to "processes" where it was used in relation to MPI.**

- Line 107+108: Either provide the name of the command line argument (if it is important), or remove the statement that it is a command line argument (because it is clear that it is activated somehow). The more interesting question here is how does a user know which "value" to choose to get a sufficiently accurate match. Is there a reasonable default value provided, or would a user have to perform manual testing to get "a feel" for a good value?

The value given to the program should be a decimal which represents the percentage change from the previous iteration (10% = 0.1), this is then used to generate a range from the previous model inside which the new model is deemed "close enough". Since this is an optional feature, we do not place a default value in the code and the user is expected to provide one when activating it. We also do not offer suggestions as to a good value, except to say that looking the full results in the tables within the appendix (currently C) the scores are relatively low and this (at least for this example) should be considered when deciding on a value. We will alter this part of the article to simplify it for the reader.

**We have removed the statement that the option is activated as a CLA and added more information about what is required by ShellSet if it is activated. We clarify for the reader that a *good* value is personal to the requirements, the underlying model, and the scoring dataset(s) used.**

- Line 111: The grid search is an interesting addition, but its description is missing crucial details about the algorithm. In particular you should spell out somewhere that your grid search is actually a contracting grid search (which searches on multiple levels, narrowing in onto the optimal results). It would also be useful to provide some context on other possible search algorithms or why you chose a grid search (see my references on the conclusion for examples). I can think of several different ways to perform this grid search, and the choice of algorithm presumable influences how well it scales in parallel. E.g. does the algorithm create one grid level, then execute all models in this level on the available cores? Then once the level is finished it creates the next level (and how? use the best value as center point in parameter space for a finer grid? or try to find the neighboring points with best values and span a grid between them?) and finishes that level before refining further? And what happens if the algorithm identifies two disjunct regions with similar error values? **Later edit**: After reading the rest of the paper I found the algorithm description in Appendix A (which is not referenced from the paper). I would suggest to move Appendix A as a subsection into Section 3 (E.g. 3.1 could be the MPI parallelism part of Section 3, 3.2 could be the grid search algorithm, 3.3. could be the user interface changes). At the very least reference Appendix A from Section 3 when you mention the grid search algorithm.

We agree that perhaps the simplest option for the reader would be to move the description of the grid search algorithm from the appendix into section 3. We will rearrange section 3 to facilitate this while avoiding having only a section 3.1.

**We have expanded section 3 in the manner suggested within the comment - including combining Appendix A with other information on the grid search into a single section 3.1. Other detail was also gathered into sections 3.2 & 3.3 related to ShellSet inherent parallelism and user interface respectively.**

• Line 120: "theoretically" and "possible" is duplicative and makes the sentence harder to understand.

This will be changed.

**Changed.**

• Line 121: Please specify the conditions that are currently implemented and checked. As a simple user it is impossible to know what "the most general conditions" are.

We will add information about the encoded variable checks performed by ShellSet.

**We have added detail about the current checks performed by ShellSet into section 3.1.**

• Line 124: It is not clear what "its variable values" refers to. Do you mean the input control parameters?

We mean the values for the altered variables - those defining the address of the model within the parameter space. We will clarify this in the text.

**The text has been rewritten to clarify the information contained in this new ShellSet output file.**

• Line 125: "improved user satisfaction" is a very general term and hard to quantify without tools like user surveys etc. What are your sources? Maybe reword to "This combination ... has reduced error-prone manual operations and saved us and our collaborators valuable research time. This allows ... ." or similar.

We will update this part in a manner similar to the suggestion.

**The text has been updated to note direct experience with the UI instead of using general terms.**

• Line 137: unnecessary "the" before "Shells"

This will be removed.

**Done.**

• Line 140: Not necessarily a comment on this manuscript but for future versions of the software: You write that you have simplified the controls of the program, but you still require the user to prepare 3 (or so?) different files and up to 9 different command line arguments. In my opinion it should be possible to combine these parameters into a single input file and maybe one or two CLAs. Some of the CLAs look like they are runtime configuration options rather than something that should go into a CLA. And one of your input files only contains paths to other input files, which seems repetetive in design (I understand that this is for historic reasons because the software controls several submodules).

As noted in the comment one of the input files is simply a list of the input files for the three component programs – this is a historical feature which was done to leave the component programs as original as possible (as in response to Daniele Melini's comment). The three new input files are actually 2: the aforementioned input file list and 1 of either a grid search setup or list of models. Of the CLAs only 2 are required for the minimum grid search setup while 1 for listed input, with others controlling optional features or dependant upon the type of model, any defaults are noted in the user guide. We have tried to design the CLAs in a way that means the most commonly changed options and optional features are simply controlled & checked at launch time instead of checking & updating an input file, however in future updates we will continue to refine the setup when needed and depending on feedback.

**No changes were required by this comment. We remain open to updates based on user feedback.**

• Line 145: This is what I was looking for before (how to compute the combined misfit), but you note that geometric mean is the new option, and you do not spell out what the old option was. Also the individual scores are still not explained. Later addition: After reading Section 4 and Bird et al. 2008 I am more confused. Bird et al 2008 already used a geometric mean to compute the combined score, how does this new method differ from the previously published one?

In the original OrbScore there were only 6 misfit scoring options. As noted in the comment the authors of Bird et al. 2008 did use the geometric mean to score their models, however it was computed manually by the authors after each simulation. In ShellSet the geometric mean is now an encoded 7th scoring option, both for the list input and to select the best models within the grid search procedure. The user can decide to include any combination of 5 misfit scoring options in the calculation of the geometric mean, those which are calculated misfits (GV, SSR, FSR, SA, SD) but not the calculated score (SC). This information will be added to the text for clarity.

**The geometric mean and which misfit scores may be used in calculating it are all clarified within the article. We have made it clear that the geometric mean needed to be calculated manually when using OrbScore but is calculated automatically by ShellSet.**

• Section 3.1 seems unnecessary as most of the important information is already given in line 35-37. Maybe include the remaining bits in the introduction (like the info about the specific oneAPI toolkits necessary) and delete this section. Also it is generally not considered good style to have a section number 3.1 if there is no 3.2. In addition it is ok to only list the dependency names in the manuscript, however the Github repository of the software should in addition list the minimum version numbers of all the dependencies (Fortran Compiler, MKL, MPI; or simply the combined Intel OneAPI version). I had to test the program on OneAPI 2024, but the authors clearly developed on some other version, so if 2024 wouldnt have worked for me I had no information on which version to use instead.

As noted, ShellSet was first developed using an older version of OneAPI and while we have had no issues using newer versions since then it is correct that version numbers be added to the GitHub repository. Section 3.1 can be summarised and moved into the introduction, while section 3 will change in line with previous comments.

**The full list of dependencies have been moved into the introduction, including potential providers. The GitHub README has been updated to state that the 2022 version of Intel oneAPI was used but that both 2023 & 2024 versions have been successfully used to compile & run the program.**

• Line 177: It is not quite clear what "the authors" is referring to (I presume the authors of the original Bird et al paper), please clarify in the text

We mean the authors of Bird et al., this will be clarified in the text.

**We have rewritten this part for clarity.**

• Line 206 - 209: This is the description of the grid search algorithm that I would have expected in a general form in Section 3. However, the description is not specific enough in its current form to answer all of my questions above. E.g. You select the best 2 models for the creation of the next grid layer, forming new 3x3 grids on the lower level. From the statement that you have 18 models on the lower models I suppose the 2 selected models do not act as corner points of the next level, but you do not spell this out. Instead I suppose you create two independent 3x3 grids around the model parameters of the best models on the coarse level by varying the parameter values from their old values, e.g. as $x_{(i+1)} = x_i +/- dx/2$ if x is a model input parameter, i the level index of the grid hierarchy, and dx the step size of the parameter variation? I also assume that the same parameter combination that was run on the coarse grid is not repeated on the finer grid? This is implied in line 209, but it would be worth spelling out more clearly (e.g. add a sentence that explains how the input parameters on the lower level grids are chosen, and that one of the models on the lower level is identical to the "parent" model on the coarser level and therefore not recomputed). **Later edit**: This comment is now obsolete after I found Appendix A, maybe you can still use some of the ideas to improve upon Appendix A. One question that is still open to me: In Appendix A you describe a 2x2 grid search in the main text a 3x3 grid search. How does the algorithm proceed in a 2x2 grid search if the central model (the one from the coarser level) is the best model? This model is not associated with any of the 4 new cells that were generated. (this is not a problem in 3x3 grids, because the original model is also a cell center on the finer level).

As noted previously we will move appendix A into section 3 and add further detail on the algorithm's workings there.

**We have moved appendix A into section 3 and added details about how the grid search works in ShellSet.**

• Fig. 1+2: The color scale label "Geometric mean" is clear to the authors, but to the reader it is not clear the geometric mean "of what" is shown here. Please reword to "Geometric mean score" or something similar. Also the tauMax axis label is missing its unit (I suspect fFric the other axis is unitless).

We will add the suggested detail to the two figures and add unit labels where necessary.

**We have altered the legend label and added tauMax units in the two figures. We have also added missing units to variables in each table in the main text and appendices where necessary.**

• Line 296: speedup performance → performance speedup

This will be changed.

**Done.**

• Line 298+299: This performance result (as well as the table) while showing some benefit of the MPI parallelization is slightly concerning. I understand the complications of the performance measurement that were also already mentioned in community comment 1 (CC1) on the discussion page of the manuscript. I also agree with the authors that the final optimal setting for most users will be to let ShellSet automatically select the number of MKL threads to optimally use the available compute cores. However, I also think the request raised in CC1 (performance table for fixed number of MKL threads) needs to be addressed. My reasoning for this is the following: The MPI problem solved by ShellSet is very close to a scenario that is called 'embarassingly parallel', which means the number of models that need to be computed are independent from each other and require minimal communication. Therefore, we would expect the compute time to scale inversely proportional to the number of available MPI worker processes as long as a sufficient number of compute cores and models to be computed are available and the number of MKL threads per model is constant. However, due to the changing number of MKL threads between the rows of table 3 this is impossible to check from the table. The only case that can be used as a test for this is the transition from 8 workers / 8 models, to 16 workers / 16 models (both of which use 1 MKL thread according to table caption), for which we would expect the compute time to remain roughly constant within some uncertainties (due to changes in model setups). However, the table clearly shows a doubling of compute time from 15 to 29.5 minutes. This implies that either: (i) the MPI ranks are not optimally distributed among the available compute cores (e.g. all ranks are always distributed among the 8 performance cores, efficiency cores are ignored), (ii) the MPI implementation is incorrect or doesnt scale beyond 8 cores, (iii) the given number of MKL threads is incorrect. A rerun of the table with a fixed MKL thread number of 1 on the same hardware (no need for HPC) could distinguish between these cases. This could prove that the authors MPI implementation is correct and the weird timing results from the complications of modern consumer CPU architecture. This would also show that running ShellSet on HPC CPUs (either workstations or HPC clusters) will be more efficient than this table can show. This is because splitting a model into more and more MKL threads will decrease the parallel efficiency, while distributing more and more models among the available MPI workers will not (on modern HPC clusters we can run ¿100,000 MPI ranks efficiently in parallel, but for the foreseeable future we will not be able to split a model in ¿100,000 MKL threads efficiently).

The comment is correct that scaling is difficult to understand from the test performed. We had tried to perform a "realistic" test (behaving how a user might in using all cores) however true program scaling performance is then hidden. We will re-run the performance test fixing the number of MKL threads to 1 in order to better understand the scaling performance of the program.

**We have split the performance analysis section into 2 parts to analyse the MKL & MPI parallel performance separately. The MKL performance has been tested on a single node of a HPC cluster while in the MPI performance section we test the performance on both *typical* desktop machine and a single node of a HPC cluster. The MPI performance analysis now shows the performance gain from removing the user interface and from running multiple models in parallel.**

• Line 305: "We have shown in Sect. 4 ..."

Will be changed.

**Done.**

• Line 306: Specify which new data set

Will be noted.

**Done.**

• Line 311: Move the project name out of the conclusion, this is what the acknowledgments section is for. Keep the future application here.

The project name is not one which funded the creation of the program but one in which the program will be used later. However, we will remove mention of the project to avoid confusion and simply state our future targets.

**We have removed the project title from the list of future works to avoid confusion about whether that project funded (or part funded) the ShellSet work.**

• Line 314-316: This statement is very vague and seems disconnected from the earlier description of the search algorithm. In particular you have not described earlier which part about the grid search algorithm is currently not efficient. I assume it is the bottleneck of choosing the next grid level after one level has been completed, therefore limiting the total number of models that can be run in parallel. The current formulation of the sentence implies you already know the algorithms you want to try ("altering the search algorithm" instead of "exploring other search algorithms"), so you should either: name the alternatives you want to explore and why (see the introduction of Baumann et al. 2014 for a list of algorithms that have been used in geodynamics and Reuber 2021 for a wider list of sampling algorithms), or at least clarify which bottlenecks exist that you have to overcome with a new algorithm.
References:
https://doi.org/10.1016/j.tecto.2014.04.037.
https://doi.org/10.1007/s13137-021-00186-y

The conclusion will be reworded for clarity, including why we expect some or another search algorithm to offer better performance on a larger machine. Combined with the detail added about the grid search in a previous section and the performance test this will hopefully clarify our reasoning to a reader. We thank the reviewer for the suggested references.

**Within section 5 we have stated why we believe the grid search algorithm to be sub-optimal compared with an ideal (potentially nonexistent) option. We have also added detail to the conclusion about potential options to update the program with additional parameter space search options.**

• Appendix A and B seem like important additions to the manuscript, leaving them to the appendix made the main paper harder to read and understand. Appendix A should certainly move into Section 3. Depending on your decision on my comment about section 2.3 Appendix B should either move into Section 2.3 or at least should be referenced from there as a new dataset available for OrbScore.

Appendix A will be merged into section 3. Since appendix B is an outline of the creation of the dataset we prefer to keep that in the appendix in order to keep the article relatively focused on the program and its performance and examples. However, we will add detail about this new (and other) datasets into section 2.3 and properly refer the reader to the appendix for detail on the creation of the new dataset.

**Appendix A was merged into section 3.1. Appendix B is now properly referenced within the article text where appropriate and more detail has been added on other scoring datasets in the OrbScore section.**

**Community comment # 1: Daniele Melini (17/11/2023)**

• As discussed in the paper, interpreting the results of ShellSet performance tests shown in Table 3 and Figure 3 is not straightforward since the code uses a hybrid MPI/OpenMP model in which the number of MKL threads is adjusted dynamically with the number of MPI workers. If technically possible, a more insightful understanding of the code scaling could be obtained by running an additional set of tests in which, for 64 models, the number of MKL threads is kept fixed. In that way it could be possible to fit the speedup curve with the Amdahl law to get an estimate of the parallel fraction of the code (which I expect that should be quite large). Moreover, by running this test with the number of MKL threads fixed to different values (for instance 1,2,4) it should also be possible to get an idea of whether in large-scale runs it is more computationally efficient to leverage the MKL parallelism or to run all the MPI workers as purely serial tasks.

I am also wondering what is the level of integration between ShellSet and its dependencies: if a new version of one of the three codes are released, how easy it is to drop it into ShellSet?

It is true that performance testing of ShellSet is a little more complicated than usual for the reasons which you have noted. Currently Intel MKL dynamically selects the number of threads based on the problem size and number of physical cores available. As noted in the article we have added a control on this to prevent an over-request of resources. We originally omitted this test since, as we state, the target for ShellSet is a 'typical', likely non-technical user who would be able to rely on the MKL dynamic option, which is set within ShellSet, to automatically select the best number of MKL threads.

One obvious drawback to this testing is that any results, while providing insight, will only be true for the specific problem size on which they are performed. In this article we tackle a global model with 16008 nodes, this is likely to have different performance scaling related to MKL thread numbers than a local model with 2000 nodes. Another is that, since ShellSet was not optimised for cluster use, the speed up potential from varying MKL threads could be partially lost within the program.

We recognise that MKL thread testing could be of interest to more technical users, and we will attempt to perform this analysis if a suitable machine can be found within publication deadlines. We note that this performance test is something which can also be provided for during future updates to the source code and user guide.

The three original codes are kept as 'whole' as possible, with only the required changes made to each one for some efficiency savings, necessary new subroutine calls, error handling etc. Also, to simplify the source code for the user we have separated the original 3 programs and their subroutines/functions into separate files. Unfortunately for these reasons it is not possible to simply exchange an updated program version for its counterpart in ShellSet. However, the vast majority of each of the 3 programs remains the same and so for typical (not root & branch) updates made to each can be simply copied into ShellSet.

Fortunately, amongst our authors we have the owner of the three programs and so in partnership will endeavour to keep ShellSet up to date with its constituent parts. It is for this reason that we would encourage users to check the ShellSet GitHub page (https://github.com/JonBMay/ShellSet) which will continue to maintain the most up to date version, while the source code available on Zenodo (https://zenodo.org/records/7986808) will be periodically updated.

**We have updated our MPI (multi model) performance tests at the request of reviewer 2. The results from these tests, performed on both a desktop computer and a high performance computing node, are shown in section 5.2. Section 5.1 shows the performance when running a single model while varying the number of parallel threads used by the MKL routines. Due to the relatively poor speedup obtained when increasing the number of MKL threads, to ensure simplicity for the reader and a certain fairness in comparing results, we have performed all tests for section 5.2 using only a single MKL thread.**

---

## Author Response (AR2)

**Editor # 1: Dr. Lutz Gross**

I am happy to inform you that your paper has been accepted subject to minor revisions. Please provide a revised manuscript that

1. address the commends of reviewer #2 - in particular about a consistent reference to the "ShellSet v1.1.0" program code.

2. includes the feedback from reviewer#1: The discussion of the performance of dgbsv is oversimplifying the situation as the performance is sensitive to many factors including memory bandwidth (as mentioned by reviewer #2), memory hierarchy, cashe size, thread placement (hyperthreading), matrix size, matrix bandwidth and competing applications (as mentioned in the manuscript). One needs to be careful to draw conclusions from the a single test case.

**We have made the changes as requested by the reviewers.**

I would like to ask you to modify the manuscripts in the following way:

2.1 better clarify how the timings were obtained (single user? single model? averaged?)

**Each of the performance tests in section 5 used the same fixed model and are averaged over 3 iterations of each test. The MKL test in section 5.1 used only a single MPI process with Shells iterated twice, the MPI test in section 5.2.2 also iterated Shells twice while the test in section 5.2.1 iterated Shells 5 times (in line with Bird et al. (2008)). Where necessary we have clarified this in the text.**

2.2 remove "this could be because the problem size is not large enough or, since we are working on a banded array, the problem is simply not complex enough." - this can not be justified by the test (meaning of 'complex'?). As using more than 16 threads is crossing CPU boundaries a possible reason could be slower cross-socket memory access and slower cross-socket thread barriers -# of thread barriers tend to increase with a smaller problem size per thread.

**We have removed the text and added a note about the slowdown possibly being due to using cores which span multiple CPUs.**

2.3 remove the paragraph after line 368: 16 independent models on 16 cores may uncoordinately compete over memory bandwidth and caches so there is no guarantee that this runs faster than a single model on 16 cores.

**We have removed this paragraph.**

**Referee #1: Dr. Lavinia Tunini**

Minor comments (# of lines referred to cleaned manuscript):

• Misleading reference. The authors, in the abstract, mention the link https://zenodo.org/records/7986808, but, in another part of the manuscript – the end of the Introduction – they cite May et al. (2023), which connects to the same Zenodo link mentioned in the abstract. In order to avoid confusions, I suggest to cite in both places, as well as in the Code and Data Availability section, the ShellSet software source as: https://zenodo.org/records/7986808, May et al. (2023).

**We have changed all references to match the suggestion.**

• The authors should specify in the manuscript (may be in the Code and Data Availability section) that May et al. (2023) is the main source for the ShellSet program v.1.1.0, and which are the main differences with respect to the version at https://github.com/JonBMay even if these differences are detailed at the github link.

**We have clarified the differences between the two repositories within the Code and Data Availability section.**

• Lines 43-44. Please, rephrase the sentence as "In this work we present ShellSet, which is composed by (i) a single program ...; (ii) and OrbScore2, a scoring program. . . ." , as well as eliminate "which we call ShellSet" at line 50.

**We have rephrased the sentence and eliminated the text at line 50.**

• Line 53. Missing dot.

**Added.**

• Line 83. Extra ")"

**Removed.**

• Line 241: "[. . . ], there are two new Python programs .."

**Done.**

**Referee #2: Dr. Rene Gassmoeller**

I recommend this article for publication, but have one remaining comment that I think would be useful to consider in the performance discussion of the final article:

The MKL thread and MPI performance tests now look much better and clearly show the speedup users can expect. For MPI you see very reasonable speedup that is almost linear up to 4-8 cores on the desktop and up to 16-32 cores on an HPC Node (let's say 6 cores on the desktop and 24 cores on the HPC node). You justify the reducing computational efficiency for larger core numbers in turn with a too small model size (for MKL threads), possible background load on the desktop (MPI desktop), or not sufficient model numbers to utilize the full HPC node (MPI HPC node). However, at least for the MPI tests, what I think more likely is that you see the transition from a compute bound to a memory-bandwidth bound algorithm. Every system has a maximum memory bandwidth that has to be shared across all MPI workers (or all MKL threads). When you increase the number of participating compute workers (threads or MPI processes) you increase the load on the memory bandwidth until the model time is no longer limited by the available compute resources, but by the speed with which new data can be delivered to the CPU cores. This is a hard limit that will be the same for different parallelization options (MPI/threads), but varies between different compute architectures (desktop vs HPC). This would also fit the observation that Core 12900 and Xeon Gold 5218 have similar maximum memory bandwidth (google search result: up to 75 GB/s for 12900, and up to 128 GB/s for 5218), but 4 Xeon Gold would have 4x the memory bandwidth (therefore increasing your scaling from $\tilde{6}$ cores to $\tilde{2}4$ cores). All of this is just a back of the envelope calculation of course since I dont know your actual memory bandwidth. However, many finite element codes are memory bandwidth limited, because they rely on sparse matrix - vector products, and short of rewriting the algorithm there is not much you can do about it (see e.g. the introduction of Kronbichler & Kormann, 2012, https://www.sciencedirect.com/science/article/pii/S0045793012001429 for a description of the problem, or Clevenger & Heister, 2021, https://onlinelibrary.wiley.com/doi/full/10.1002/nla.2375 for an implementation of a matrix-free algorithm for the Stokes equation that circumvents the memory bandwidth limit).

• I think it would be useful to mention the possibility of a memory bandwidth limitation somewhere in the performance discussion, because it gives a uniform explanation of the scalability limit you see. It also means that your scalability is not inherently limited to the speedup you observe, because if you increase the number of models further and distribute them across several HPC nodes you will see further speedup (each node increases the available memory bandwidth), therefore making the case that ShellSet can indeed be used for much larger studies (say ¿1,000 models) as long as sufficient HPC resources are available.

**We have noted that memory bandwidth may be a cause of reduced performance increase, and stabilising, within the performance section. We have also added reference to this inside the conclusion.**